# MOMENTUM-DRIVEN NOISE-FREE GUIDED CONDITIONAL SAMPLING FOR DENOISING DIFFUSION PROBABLISTIC MODELS

## ABSTRACT

We present a novel approach for conditional sampling of denoising diffusion probabilistic models (DDPM) using noise-free guidance, which eliminates the need of noise-finetuning, and can be applied to a wide range of guidance functions operating on clean data. We observe that the performance gap between previous clean estimation $(\widehat{x_0})$-based methods and noised sample $(x_t)$-based methods stems from the incorporation of estimation deviation in the clean-estimation guidance process. The former contrasts with noise-guided techniques where noise contamination is addressed by a noise-finetuned classifier, leading to inconsistent and unreliable guidance gradients from the inaccurate clean estimation. To tackle this issue, we propose a two-fold solution: (1) implementing momentum-driven gradient filtering to stabilize the gradient transmitted from the guidance function, ensuring coherence throughout the denoising process, and adaptively adjusting the update stepsize of pivot pixels to increase their resilience against detrimental gradients; and (2) introducing a guidance suppression scheme to alleviate the impact of unreasonably large weights assigned considering the significantly larger estimation deviation in the early stage. Extensive experiments demonstrate the superiority of our method on clean guided conditional image generation. Moreover, our method offers the potential for reusing guidance on DDPM with other noise schedules and we apply it to the arbitrary style transfer task, achieving state-of-the-art performance without being limited to labeled datasets.

## 1 INTRODUCTION

Recent years have witnessed a flourish of deep generative models. With the help of powerful architectures such as GAN (Goodfellow et al., 2020; Karras et al., 2019), VAE (Kingma & Welling, 2013; Vahdat & Kautz, 2020), and autoregressive models (Child et al., 2019), machines can learn from existing real-world data and generate realistic images (Razavi et al., 2019), natural language (Brown et al., 2020), and audio (Oord et al., 2016). Recently, a new class of generative models, denoising diffusion probabilistic models (DDPM) (Ho et al., 2020), has started to make a splash across the generation fields and has achieved state-of-the-art performance in many downstream applications (Luo & Hu, 2021; Ramesh et al., 2022; Preechakul et al., 2022; Saharia et al., 2022c). On a Markov chain, DDPM gains its generative power by adding noise to clean data and learning the reverse process to denoise and generate new data. Compared with traditional generative models, it is training-stable and allows for better generative diversity with higher fidelity and richer generative details.

In order to generate data with desired semantics using DDPM, great efforts have been made for conditional sampling. Mainstream approaches can be broadly classified into four categories: 1) The first kind (Dhariwal & Nichol, 2021; Song et al., 2020b) proposes to use the gradients of noised guidance functions (*e.g.* noised classifiers or noised CLIP) for conditional sampling. However, it introduces additional costs for training or finetuning the guidance functions on levels of noised data, and is limited to only labeled datasets (Dhariwal & Nichol, 2021). 2) The second kind employs classifier-free guidance (Ho & Salimans, 2022; Ramesh et al., 2022; Rombach et al., 2022b), which performs implicitly guided sampling with the difference between the conditional and unconditional scores to remove the extra classifier. This approach merges the separate training phases for classifiers and diffusion models into one, while it doubles the sampling cost. Meng et al. (2023) proposes to

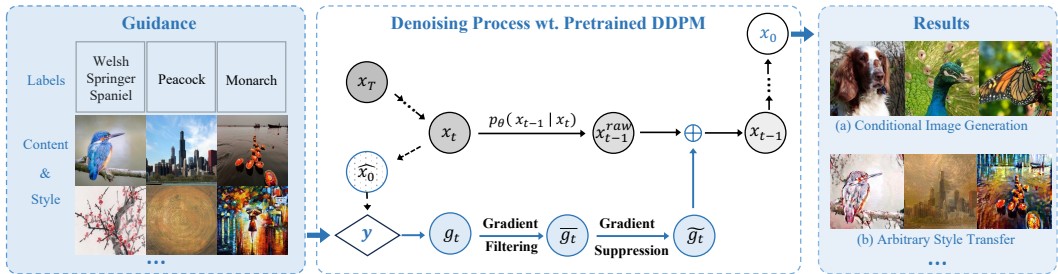

Figure 1: Given arbitrary conditions on clean samples, our method guides diffusion models throughout the denoising process using the clean estimation as a proxy. Gradient filtering and gradient suppression schemes are proposed to solve the estimation deviation problems during the guidance process.

distill classifier-free guided diffusion model to reduce the sampling cost. However, it is still limited to paired data, and requires retraining the entire diffusion model, rather than noised classifiers only, for different types of guidance, thus placing extremely high demands on computational resources and causing unavoidable inconveniences. 3) For the third kind, conditional sampling is achieved through outline-based guidance, such as using strokes (Meng et al., 2021) or low-frequency image component (Choi et al., 2021) as references for a conditional generation. Unfortunately, these methods can only provide a rough guidance direction, limiting deeper and broader control over the generated results. 4) The last kind (Kim et al., 2022) achieves conditional generation by finetuning the diffusion model according to semantic requirements. This method requires frequent finetuning for different inputs, imposing a computational burden.

Thus a natural question arises: can conditional sampling be achieved in a more general, lightweight, and efficient way? Recently, Graikos et al. (2022) make a similar attempt and propose to treat DDPM as plug-and-play priors. They formulate the generation process as a stochastic optimization problem, constraining intermediate steps not to stray too far from the DDPM process while maximizing its likelihood using a clean guidance function only. However, due to its approximation for the distribution dependency between $p(x_t|x_0)$ and $p(x_{t-1}|x_0)$, their approach is difficult to generate realistic samples on unaligned datasets like ImageNet. At the same time, other works (Crowson; Avrahami et al., 2022) explore to leverage the clean estimation $\widehat{x_0}$ for guidance. Although they are noise-training free, they do not achieve comparable results to previous noise-guidance-based methods (Kim et al., 2022). Yu et al. (2023) and Bansal et al. (2023) intend to improve the fidelity of clean guided DDPM samples with time-travel strategies. However, this approach increases the sampling cost by multiple times and leads to sub-optimal overall sampling quality due to insufficient data coverage.

Our key observation reveals that the estimation deviation of $\widehat{x_0}$ is incorporated into the original clean-estimation guidance process, which contrasts with the noise-guided methods where noise contamination on $x_t$ is addressed by a noise-finetuned classifier. Consequently, the guidance gradient from the inaccurate $\widehat{x_0}$ is not entirely consistent and reliable, resulting in suboptimal outcomes. Drawing inspiration from neural network training optimizers, we propose a ***momentum-driven gradient filtering*** approach for clean estimation guided conditional DDPM. Specifically, we employ first- and second-order momentum to stabilize the gradient transmitted from the guidance function, ensuring coherence throughout the denoising process, and adaptively adjust the update stepsize of pivot pixels to increase their resilience against detrimental gradients. Furthermore, we identify that in original clean-estimation-based techniques, the early gradient is assigned unreasonable large weights considering the substantially larger estimation deviation in the early stage. By rectifying this issue using our proposed ***gradient suppression*** scheme, performance can be further enhanced. As shown in Fig. 1, our method is simple and effective, and extensive experimental results demonstrate its superiority on boosting the performance of downstream tasks.

In the following, we summarize our main contributions:

1) We introduce the momentum-driven filtering and the gradient suppression scheme to the clean estimation guided conditional sampling for DDPM.

2) Our method utilizes clean guidance functions, eliminating the need for additional training. Moreover, it is versatile and applicable to a wide array of guidance functions on clean data and is not limited to labeled datasets.

3) We demonstrate the feasibility and generalization capability of our framework with state-of-the-art performance on several downstream tasks.

## 2 RELATED WORK

### 2.1 DENOISING DIFFUSION PROBABILISTIC MODELS

DDPM represents a new class of generative models that offer better generation quality and diversity compared to traditional methods like Generative Adversarial Networks (GAN)(Goodfellow et al., 2020; Brock et al., 2018b; Karras et al., 2019) and Variational Autoencoder (VAE)(Kingma & Welling, 2013; Razavi et al., 2019; Vahdat & Kautz, 2020). Song et al. (2020a) introduced the Denoising Diffusion Implicit Model (DDIM), which connects with score matching and inspires subsequent sampling acceleration research (Bao et al., 2022a; Lu et al., 2022; Karras et al., 2022; Watson et al., 2021; Liu et al., 2022). Efforts have also been made to enhance model generation capabilities, such as learnable variance, cosine noise schemes and model architecture improvements by Dhariwal & Nichol (2021), and generalizing the Gaussian noise schedule of DDPM to various degradation schedules by Bansal et al. (2022). Daras et al. (2022) introduced momentum for unconditional generation, signifying a focus on a consistent and stable generation process. DDPM has been used in numerous downstream tasks such as text-to-image generation (Ramesh et al., 2022; Saharia et al., 2022a; Rombach et al., 2022b), image super-resolution (Choi et al., 2021; Li et al., 2022a; Saharia et al., 2022b), 3D point cloud generation (Luo & Hu, 2021), speech and text generation Chen et al. (2020); Austin et al. (2021), image in-painting (Lugmayr et al., 2022; Song et al., 2020b), and video generation (Ho et al., 2022a; Yang et al., 2022; Ho et al., 2022b).

### 2.2 CONDITIONAL SAMPLING FOR DDPM

Conditional sampling aims to generate data with desired semantics. Dhariwal & Nichol (2021) employ gradient-based noised classifier guidance, enhancing DDPM's generative capabilities, but with added computational overhead for training classifiers on noised data. Ho & Salimans (2022) introduce classifier-free guidance, removing extra classifiers and inferring the implicit guidance from the gap between the conditional and unconditional predictions (Ramesh et al., 2022), enabling large-scale model training (Ramesh et al., 2022; Nichol et al., 2021; Saharia et al., 2022a). However, this approach doubles the sampling cost and supports only specific guidance types, with both methods constrained to paired data (Dhariwal & Nichol, 2021). Our proposed method overcomes these limitations, offering a new conditional sampling paradigm with reduced training burden.

Meng et al. (2021) and Choi et al. (2021) investigate lightweight conditional sampling methods based on stroke and low-frequency components, but offer limited control over results. Graikos et al. (2022) employ diffusion models as plug-and-play priors. Crowson explore clean estimation-based conditional sampling, but their outcomes are not as comparable to the noise-guidance-based methods. Both Yu et al. (2023) and Bansal et al. (2023) attempt to improve clean-estimation guidance using time-travel strategies, inccuring multiple times more sampling cost. Our approach leverages momentum-driven gradient filtering and gradient suppression mechanisms to improve clean-estimation guided DDPM with no extra computation cost.

### 2.3 STYLE TRANSFER

Style transfer is a classic and influential task in image generation, with seminal works like NST (Gatys et al., 2016) and AdaIN (Huang & Belongie, 2017) impacting other fields (Karras et al., 2019; Park et al., 2019; Choi et al., 2020; Zheng et al., 2019). The task can be categorized into single model single style (SMSS)(Johnson et al., 2016a; Ulyanov et al., 2016b;a), single model multiple styles (SMMS)(Chen et al., 2017; Dumoulin et al., 2016), and single model arbitrary styles (SMAS)(Ghiasi et al., 2017; Huang & Belongie, 2017; Park & Lee, 2019; Li et al., 2017b; Gatys et al., 2016; Chen et al., 2021). The third category, also known as arbitrary or universal style transfer, is the most challenging and our primary focus. Gatys et al. (2016) first apply neural networks to this task, using Gram matrices of VGGNet-generated feature maps (Simonyan & Zisserman, 2014) to represent image styles. Huang & Belongie (2017) propose the feed-forward AdaIN method, aligning mean and variance of feature maps, while Li et al. (2017b) match Gram matrices of content and style images for style transfer. Subsequent works (Deng et al., 2021a; Yao et al., 2019; Liu et al., 2021a; Wang et al., 2020; Chen et al., 2021; Lu & Wang, 2022) improve transfer consistency and quality, but generating realistic results remains challenging. Arbitrary style transfer, a classic task on unlabeled datasets, brings inherent difficulties for previous DDPM methods. However, our approach effectively addresses arbitrary style transfer, demonstrating potential for similar downstream tasks.

## 3 BACKGROUND

**Unconditional Sampling**   For a data distribution $x_0 \sim q(x_0)$, the forward process progressively adds Gaussian noise to it until it converges to isotropic Gaussian $x_T \sim \mathcal{N}(0, \mathbf{I})$, given large enough T and suitable noise schedule $\beta_t$. The noised sample $x_t$ can be obtained from the Markov Chain:

$$q(x_t|x_{t-1}) = \mathcal{N}(x_t; \sqrt{1-\beta_t}x_{t-1}, \beta_t\mathbf{I}), \tag{1}$$

$$x_t = \sqrt{1-\beta_t}x_{t-1} + \sqrt{\beta_t}\epsilon_t, \tag{2}$$

or directly conditioned on the clean data $x_0$, with $\alpha_t = 1 - \beta_t$ and $\bar{\alpha}_t = \Pi_{i=1}^{t}\alpha_i$:

$$q(x_t|x_0) = \mathcal{N}(x_t; \sqrt{\bar{\alpha}_t}x_0, (1-\bar{\alpha}_t)\mathbf{I}), \tag{3}$$

$$x_t = \sqrt{\bar{\alpha}_t}x_0 + \sqrt{1-\bar{\alpha}_t}\bar{\epsilon}_t. \tag{4}$$

In the reverse process, DDPM is trained to learn a parameterized Gaussian transition $p_\theta(x_{t-1}|x_t)$ to approximate the posterior $q(x_{t-1}|x_t, x_0)$ given by the Bayes Theorem. New samples can be generated by iteratively denoising the random noise $x_T \sim q(x_T)$ with:

$$p_\theta(x_{t-1}|x_t) = \mathcal{N}(x_{t-1}; \widetilde{\mu}_\theta(x_t, t), \Sigma_\theta(x_t, t)), \tag{5}$$

where $\widetilde{\mu}_\theta(x_t, t) = \frac{1}{\sqrt{\alpha_t}}(x_t - \frac{1-\alpha_t}{\sqrt{1-\bar{\alpha}_t}}\epsilon_\theta(x_t, t))$ and the variance $\Sigma_\theta(x_t, t)$ can be fixed as $\beta_t\mathbf{I}$ or $\frac{(1-\bar{\alpha}_{t-1})\beta_t}{1-\bar{\alpha}_t}\mathbf{I}$ (Ho et al., 2020), or learned by neural networks (Nichol & Dhariwal, 2021).

**Conditional Sampling with Noised Guidance**   Given the powerful generative ability of DDPM, it is natural to explore how to turn it from an unconditional model $p(x_0)$ to a conditional one $p(x_0|y)$. Taking $p_\theta(x_{t-1}|x_t, y)$ as the conditional reverse process, the conditional distribution follows:

$$p_\theta(x_{0:T}|y) = p(x_T)\prod_{i=1}^{T}p_\theta(x_{t-1}|x_t, y). \tag{6}$$

Existing works bridge the conditional sampling with the unconditional sampling using:

$$p_{\theta,\phi_t}(x_{t-1}|x_t, y) = \mathcal{Z}p_\theta(x_{t-1}|x_t)p_{\phi_t}(y|x_{t-1}), \tag{7}$$

where $\mathcal{Z}$ is a normalizing constant, and $p_{\phi_t}(y|x_{t-1})$ represents the guidance function given $x_{t-1}$.

In order to obtain $p_{\phi_t}(y|x_{t-1})$, some previous works resort to extra noised neural networks, for example, classifiers $p_{\phi_t}(y|x_t)$ on noised images (Dhariwal & Nichol, 2021; Song et al., 2020b) or noise-finetuned CLIP (Liu et al., 2021b). By using gradients of the noised neural network, the conditional sampling with guidance can be formulated as:

$$\widetilde{\mu}_{\theta,t} \leftarrow \widetilde{\mu}_{\theta,t} + s\Sigma_{\theta,t}\nabla_{x_t}log(p_{\phi_t}(y|x_t)), \tag{8}$$

where $\widetilde{\mu}_{\theta,t}$ is the Gaussian mean and $s$ is a scaling factor for guidance gradients.

## 4 METHODS

Noised-sample-based guidance requires training of an extra time-dependent model $p_\phi(y|x_t, t)$ on the same noise schedule as the diffusion model. In contrast, applying guidance on clean estimation(Bansal et al., 2023; Yu et al., 2023) seamlessly incorporates any off-the-shelf model and broadens the range of applicable guidance conditions to include nearly any objective function. However, these advantages do not inherently yield performance comparable to the guidance on noised samples. We propose the following plug-and-play methods that improve the performance of clean-estimation guidance. We first outline the clean-estimation guidance approach in Sec. 4.1. Next, we present our key observations on clean-estimation guidance that impact its performance and introduce our momentum-driven gradient filtering approach and gradient suppression scheme in Sec. 4.2 and Sec. 4.3, respectively.

### 4.1 NOISE-FREE GUIDANCE VIA CLEAN ESTIMATION

When sampling from a pretrained DDPM $\bar{\epsilon}_{\theta,t}(x_t, t)$, a straightforward approach for applying guidance on noised sample $x_t$ would be bearing extra costs to train a noise-robust model. Alternatively, a more

flexible and efficient approach is taking the clean estimation $\widehat{x_0}(x_t, t)$ as a proxy for noised samples. Derived from Eq.4, the clean estimation for a noised sample $x_t$ can be computed in one step as:

$$\widehat{x_0}(x_t, t) = \frac{1}{\sqrt{\bar{\alpha}_t}}(x_t - \sqrt{1 - \bar{\alpha}_t}\bar{\epsilon}_{\theta,t}(x_t, t)). \tag{9}$$

Instead of training a separate model to learn the log probability $\log p_t(y|x_t)$ of noised samples, we can utilize the widely-used objective functions on clean data $\mathcal{P}(\cdot)$ as guidance since they are now conditioned on the clean samples. Similar to classifier guidance formulated as Eq. 8, guidance via clean estimation can be achieved by:

$$\tilde{\mu}_{\theta,t} \leftarrow \tilde{\mu}_{\theta,t} + s\Sigma_{\theta,t}\nabla_{x_t}log(\mathcal{P}(y, \widehat{x_0}(x_t, t))), \tag{10}$$

where $y$ represents a wide range of guidance functions applied to the regular clean samples. The sole constraint on the guidance function $\mathcal{P}$, which serves as an implicit premise for the proof of the classifier guidance, is the consistency with a probability density function. For regular loss functions that operate on clean samples, this requirement can be readily met using a negative exponential mapping function. In the case of latent diffusion models (Rombach et al., 2022a), we integrate the latent decoder into the guidance function to map the latent features to the pixel space before applying the guidance on the clean images.

Although the clean-estimation guidance appears ingenious and practical thus far, it typically demonstrates inferior sampling results compared to the noised guidance. We discovered that the primary reason is that the guidance gradient is negatively affected by inaccurate clean estimations, particularly those derived from the highly noised samples during the early stage of the denoising process.

## 4.2 MOMENTUM-DRIVEN GRADIENT FILTERING

The gradient of the guidance function on the clean estimation continues to direct the denoising process of the noised samples, as ensured by the chain rule:

$$\frac{\partial \log(\mathcal{P}(y, \widehat{x_0}))}{\partial x_t} = \frac{\partial \log(\mathcal{P}(y, \widehat{x_0}))}{\partial \widehat{x_0}} \cdot \frac{\partial \widehat{x_0}}{\partial x_t}, \tag{11}$$

which contains two terms. The first term is the partial derivatives of the log probability w.r.t the clean estimation $\widehat{x_0}$. The second term can be written as $\frac{\partial \widehat{x_0}}{\partial x_t} = \frac{1}{\sqrt{\bar{\alpha}_t}}\left(1 - \sqrt{1 - \bar{\alpha}_t}\frac{\partial \bar{\epsilon}_{\theta,t}(x_t, t)}{\partial x_t}\right)$, which only depends on the DDPM model. Due to the inaccurate clean estimation, the first term is relatively unstable during the denoising process. The noisy guidance can be confirmed by evaluating the guidance gradients during the denoising process, as shown in Fig. 2(a). With only a clean classifier, the guidance function $\mathcal{P}(y, \widehat{x_0})$ proves to be less reliable than the noise-robust classifier(Dhariwal & Nichol, 2021). As a consequence, it can generate inaccurate guidance at certain timesteps, ultimately leading to the degradation of the sample quality.

Base on this fact, our objective is to identify a method capable of filtering the noise present in the gradient. An intuitive approach is the momentum algorithms that are heavily used in the optimizers(Kingma & Ba, 2014). With $m_t = \eta_m \cdot m_{t+1} + (1 - \eta_m) \cdot g_t$, $v_t = \eta_v \cdot v_{t+1} + (1 - \eta_v) \cdot g_t^2$ being the first- and second-order momentum of the historical gradient from $T$ to $t + 1$, we define our momentum-driven gradient filtering as:

$$\bar{g}_t = \lambda \cdot \frac{m_t/(1 - \eta_m^{T-t+1})}{\sqrt{v_t/(1 - \eta_v^{T-t+1})} + \varepsilon}, \tag{12}$$

where the $\eta_m$, $\eta_v$ are the weighting parameters of the momentum algorithms and $\lambda$ indicates the learning rate. More implementation details can be found in Appendix C.1.

As the filtering scheme involves both first- and second-order momentum, the impact of our Momentum-driven Gradient Filtering approach is two-fold. As Fig. 2(b) shows, the first-order momentum stabilizes the gradient by computing an exponentially weighted sum of the historical gradients, making it less noisy and more robust to the inaccurate estimation of the clean samples. The second-order momentum adaptively adjusts the update stepsize of each pixel according to the history of the squared gradients. Throughout the sampling process, foreground pixels, which are more sensitive to the guidance conditions, tend to be updated more frequently by the guidance

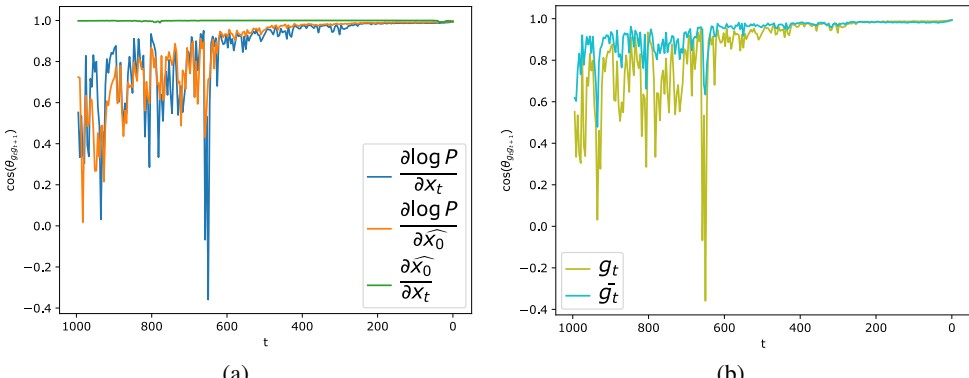

(a)          (b)

Figure 2: The cosine value of the angle between the gradient vector at timestep $t$ and that at timestep $t + 1$. The lower the curve, the noisier the gradient direction. (a) shows the guidance gradient via clean-estimation (blue) and its first term (orange), which is related to clean-estimation and relatively unstable, causing noisy guidance. (b) shows the unprocessed gradient $g_t$ given by the clean-estimation guidance and the filtered gradient $\bar{g}_t$ by our proposed method. More samples are in Appendix C.2.

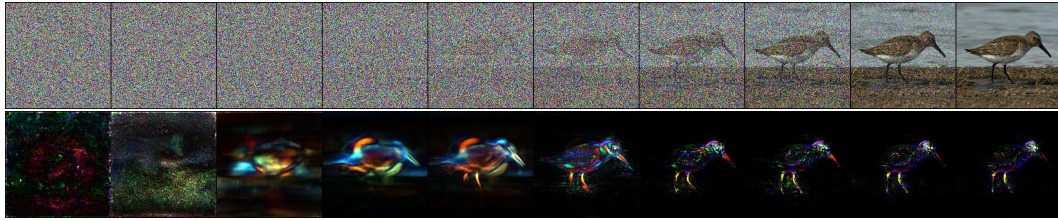

Figure 3: A randomly generated sample guided by the filtered gradient. We present its denoising process (upper) and the corresponding second-order momentum (lower). More are in Appendix C.3.

gradients. These pixels are assigned with smaller learning rates to prevent them from being easily misled by occasional inaccurate gradients. For instance, during the conditional generation process of the image depicted in Fig. 3, the pixels exhibiting larger second-order momentum (i.e., smaller stepsize) predominantly correspond to the foreground pixels on the sandpiper.

### 4.3 GRADIENT SUPPRESSION

As we observed, the norm of the guidance gradient is substantially larger in the early stage than in the late stage. This effect is undesirable based on the subsequent derivation regarding the error of the clean estimation.

The perfect clean estimation $x_0$ of a noised sample $x_t$ can be computed with the true noise $\bar{z}_t$ added during the forward process, as follow:

$$x_0 = \frac{1}{\sqrt{\bar{\alpha}_t}}(x_t - \sqrt{1 - \bar{\alpha}_t}\bar{z}_t). \quad (13)$$

Then the prediction error between the actual prediction $\hat{x}_0 = \frac{1}{\sqrt{\bar{\alpha}_t}}(x_t - \sqrt{1 - \bar{\alpha}_t}\bar{\epsilon}_\theta(x_t, t))$ and the prefect estimation is

$$x_0 - \hat{x}_0 = \sqrt{\frac{1}{\bar{\alpha}_t} - 1}(\bar{\epsilon}_\theta(x_t, t) - \bar{z}_t). \quad (14)$$

It is reasonable to regard the $(\bar{\epsilon}_\theta(x_t, t) - \bar{z}_t)$ term as well-bounded, given it corresponds to the prediction error of a 0-1 Gaussian noise vector by a well-trained DDPM model. Thus, the prediction

---

**Algorithm 1** Momentum-driven Noise-free guided sampling, given a pretrained diffusion model $\bar{\epsilon}_{t,\theta}(x_t, t)$ and a clean guidance function $\mathcal{P}$.

**Input:** Conditions $y_i$, gradient filter $F$
**Output:** Generated sample $x_0$
1: Sample $x_T \sim \mathcal{N}(0, I)$
2: **for all** $t$ from $T$ to 1 **do**
3:    $\tilde{\mu}_{\theta,t} \leftarrow \frac{1}{\sqrt{\bar{\alpha}_t}}(x_t - \frac{1-\alpha_t}{\sqrt{1-\bar{\alpha}_t}}\bar{\epsilon}_{t,\theta}(x_t, t))$
4:    $\Sigma_{\theta,t} \leftarrow \frac{(1-\bar{\alpha}_{t-1})(1-\alpha_t)}{1-\bar{\alpha}_t}I$ or network prediction
5:    $\widehat{x_0} \leftarrow \frac{1}{\sqrt{\bar{\alpha}_t}}(x_t - \sqrt{1 - \bar{\alpha}_t}\bar{\epsilon}_{t,\theta}(x_t, t))$
6:    $g_t \leftarrow \nabla_{x_t}\log(\mathcal{P}(y_i, \widehat{x_0}))$
7:    update $m_t$ and $v_t$
8:    $\bar{g}_t \leftarrow F(\lambda, g_t, m_t, v_t)$
9:    $\tilde{g}_t \leftarrow \text{Suppression}(\bar{g}_t, t)$
10:    $x_{t-1} \leftarrow$ sampled from $\mathcal{N}(\tilde{\mu}_{\theta,t} + s\Sigma_{\theta,t}\tilde{g}_t, \Sigma_{\theta,t})$
11: **end for**
12: **return** $x_0$

---

error of the clean estimation mainly depends on the coefficient $\sqrt{\frac{1}{\bar{\alpha}_t} - 1}$. The prediction error is

much larger at the early stage of the denoising process than in the later stage. However, the unreliable gradient in the early stage is assigned with unfairly large weights in the denoising process. It implies the necessity to further suppress the gradient norm on the early stages even though it has already been filtered. Illustrations and more analysis are shown in the supplementary. We thus suppress the early gradient with a simple but effective linear scheme with hyperparameter k of suppression level:

$$\tilde{g}_t = (1 - \frac{t}{kT})\bar{g}_t. \tag{15}$$

We conclude our improved clean-estimation guided sampling method as Algorithm. 1.

## 5 EXPERIMENTS

We first apply our proposed methods on clean-estimation guided conditional image generation on ImageNet (Deng et al., 2009) to demonstrate the improvements achieved on sample quality. Moreover, we guide the DDPM model to achieve arbitrary style transfer, a task unattainable by classifier guidance, by capitalizing on the flexibility offered by clean-estimation guidance. Lastly, we conduct a series of ablations on both tasks to verify the effectiveness of each part.

### 5.1 CLEAN GUIDED CONDITIONAL IMAGE GENERATION

**Setup** We evaluate our proposed methods with two strong diffusion backbone, ADM (Dhariwal & Nichol, 2021) and DiT (Peebles & Xie, 2022). For fair comparison, we take the same UNet encoder in Dhariwal & Nichol (2021) with a fixed time-embedding at t=0 to serve as the pretrained clean classifier. Following previous works (Dhariwal & Nichol, 2021; Li et al., 2022b; Song et al., 2020b), we evaluate the generation quality comprehensively using FID(Heusel et al., 2017), sFID(Nash et al., 2021), precision, and recall (Kynkäänniemi et al., 2019), where FID and sFID are primary evaluation metrics and precision and recall are secondary metrics. Additional details are in Appendix A.1.

Table 1: Comparison with exisiting methods on improving clean-estimation guidance on ImageNet (256 × 256). All results are evaluated using the TensorFlow suite from Dhariwal & Nichol (2021) on 50K samples sampled with 250 DDPM steps. †Sampled with a pretrained noised classifier with a fixed time-embedding $t = 0$. ‡Sampled with a clean classifier trained on clean samples, with less than 2% seen samples for training the noised one.

| Model | FID↓ | sFID↓ | Prec↑ | Rec↑ |
|---|---|---|---|---|
| DiT + raw clean guidance | 3.54 | **5.22** | 0.80 | 0.56 |
| DiT + Plug-and-Play (Graikos et al., 2022) | 182.59 | 279.56 | 0.10 | 0.08 |
| DiT + ED-DPM (Li et al., 2022b) | 4.41 | 5.41 | **0.84** | 0.50 |
| **DiT + Ours†** | **3.46** | 5.31 | 0.79 | **0.57** |
| ADM + raw clean guidance | 4.99 | 5.58 | 0.83 | 0.51 |
| ADM + FreeDoM (Yu et al., 2023) | 8.66 | 6.84 | **0.90** | 0.35 |
| ADM + Plug-and-Play (Graikos et al., 2022) | 117.01 | 34.17 | 0.23 | 0.20 |
| ADM + ED-DPM (Li et al., 2022b) | 5.98 | 5.93 | 0.87 | 0.42 |
| **ADM + Ours†** | **4.20** | 5.17 | 0.82 | 0.52 |
| **ADM + Ours‡** | 4.21 | **4.94** | 0.80 | **0.53** |

**Comparison** Our proposed methods primarily focus on improving the sampling quality of clean-estimation guidance technique. Plug-and-Play (Graikos et al., 2022) generates conditional samples from white noise using conditional diffusion models as priors and clean classifiers as constraints, yet struggling to generate realistic samples on unaligned datasets like ImageNet due to theoretical approximations. FreeDoM (Yu et al., 2023) and Universal Guidance (Bansal et al., 2023) both intend to improve the clean-estimation guidance with time-travel strategies. Nevertheless, these approaches incur a significantly higher sampling cost and tend to over-amplify the guidance signal for conditional image generation, resulting in high fidelity but low diversity. ED-DPM (Li et al., 2022b) dynamically amplifies the guidance gradient to avoid gradient vanishing of the noised classifier. It shows sub-optimal results when applied to clean guidance, as the EDS process can occasionally amplify inaccurate gradients stemming from low-entropy predictions on incorrect clean-estimations.

As shown in Tab.A.5, our method achieves the best overall sampling quality of clean-guided diffusion models with no additional training costs, compared to all other methods. Most importantly, our proposed methods enable clean-estimation guided ADM to outperform noised-guided one on both

Table 2: The average metrics of inputs and stylized results of different methods. $\mathcal{L}_{content}$ and $\mathcal{L}_{style}$ are calculated using a pretrained VGGNet to get perception measurement on the stylization quality.

| Metrics | Input | Ours | NST | AdaIN | WCT | Linear | AAMS | MCCNet | ReReVST | AdaAttN | IECAST | CSBNet | AesPA |
|---|---|---|---|---|---|---|---|---|---|---|---|---|---|
| $\mathcal{L}_{content} \downarrow$ | 0.00 | **4.70** | 8.39 | 7.37 | 14.60 | 5.63 | 8.56 | 8.18 | 5.55 | 8.14 | 6.48 | 6.08 | 7.66 |
| $\mathcal{L}_{style} \downarrow$ | 16.14 | **1.54** | 2.15 | 4.35 | 2.75 | 4.95 | 7.32 | 2.84 | 5.68 | 3.43 | 7.25 | 3.33 | 5.25 |

FID (4.20 vs. 4.59) and sFID (5.17 vs. 5.25) while maintaining comparable precision and recall. To ensure fair comparisons with prior works employing a noised classifier, we utilize their identical pretrained noised classifier with a fixed time-embedding input $t = 0$ as a clean version. Comparable results are attainable by training a clean classifier from scratch, requiring less than 2% (128 vs. 2.4 million samples seen by each classifier) of the training cost compared to a noise-finetuned variant.

## 5.2 ARBITRARY STYLE TRANSFER

In our approach, the guidance function operates on clean estimations, thereby enabling the application of a broad range of objective functions as guidance conditions. This advancement unlocks a plethora of novel opportunities for employing DDPM across various tasks and applications, encompassing, but not limited to, inpainting, colorization, and semantic synthesis. We explore this feature by applying guided diffusion models to the single model arbitrary style transfer task, which is challenging to address using previous methods with DDPM.

Given style hint $I_s$, arbitrary style transfer aims to migrate arbitrary styles from $I_s$ to content images $I_c$. It stands for a huge class of guidance function $\mathcal{P}$ that is hard to anticipate using label-based classifiers. Huang et al. (Huang & Belongie, 2017) use the mean and variance of feature maps generated by a pretrained VGGNet (Simonyan & Zisserman, 2014) to represent the style information. The differences between the content image $I_c$ and the stylized image $I_{cs}$ is expressed as:

$$\mathcal{L}_{content} = \sum_i ||\phi_i(I_{cs}) - \phi_i(I_c)||_2 \tag{16}$$

where $\phi_i(\cdot)$ denotes the feature map from the $i^{th}$ layer in VGGNet. Meanwhile, the style differences against the style image $I_s$ can be denoted as:

$$\mathcal{L}_{style} = \sum_i ||\mu(\phi_i(I_{cs})) - \mu(\phi_i(I_s))||_2 + \sum_i ||\sigma(\phi_i(I_{cs})) - \sigma(\phi_i(I_s))||_2 \tag{17}$$

where $\mu(\cdot)$ and $\sigma(\cdot)$ represent channel-wise mean and variance of the feature maps. Most existing works only rely on the learned content and style priors from the pretrained VGGNet (Simonyan & Zisserman, 2014), leading to unnatural artifacts. To incorporate prior knowledge of real artworks into style transfer, we utilize the pretrained unconditional DDPM on the WikiArt (Phillips & Mackintosh, 2011) dataset. Then, We map the mixture of $\mathcal{L}_{content}$ and $\mathcal{L}_{style}$ to a p.d.f.-like function as the guidance for conditional image generation: $\mathcal{P} = \mathcal{Z}e^{-(\lambda_c \mathcal{L}_{content} + \lambda_s \mathcal{L}_{style})}$, where $\mathcal{Z}$ is a normalizing constant and the ratio $\lambda_c : \lambda_s$ denotes the weights for content and style loss in the guidance function.

**Setup** An unconditional ADM was trained on the WikiArt dataset (Phillips & Mackintosh, 2011), utilizing a learning rate of $10^{-4}$ and a batch size of 8 for 1.2 million iterations with a resolution of $256^2$. The network architecture and other settings adhere to the work conducted by Dhariwal et al.(Dhariwal & Nichol, 2021). Furthermore, following the methodology presented in AdaIN(Huang & Belongie, 2017), a pretrained VGG-19 model (Simonyan & Zisserman, 2014) was employed to supply content and style guidance. Additional details can be found in Appendix.

**Comparison** We compare our method with several representative methods for arbitrary style transfer, including NST (Gatys et al., 2016), AdaIN (Huang & Belongie, 2017), WCT (Li et al., 2017b), Linear (Li et al., 2019), AAMS (Yao et al., 2019), MCCNet (Deng et al., 2021a), ReReVST (Wang et al., 2020), AdaAttN (Liu et al., 2021a), IECAST (Chen et al., 2021), CSBNet (Lu & Wang, 2022) and AesPA-Net (Hong et al., 2023). In line with many previous works (Deng et al., 2021b; Lu & Wang, 2022; xin Zhang et al., 2022), we employ perceptual errors as the metric for assessing the stylized quality of the generated output. A total of 28 content images and 62 style images are randomly selected, resulting in the generation of 1,736 stylized images. Subsequently, we compute the average perceptual content error ($\mathcal{L}_{content}$) and style error ($\mathcal{L}_{style}$) for the generated images, with respect to the content and style images. As evidenced in Tab. 2, the stylized results produced by our improved clean-estimation guided DDPM significantly outperform all other methods by a considerable margin, highlighting the immense potential of proposed techniques.

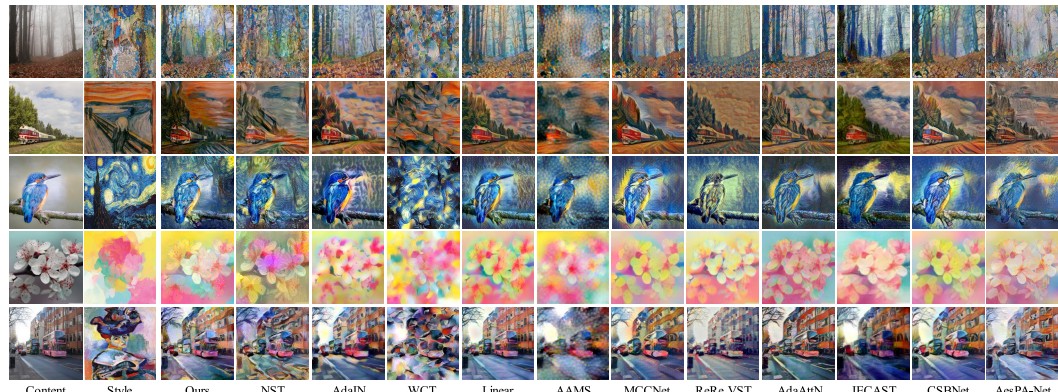

Figure 4: Qualitative comparisons of arbitrary style transfer methods.

For qualitative comparisons in Fig.4, our method can generate more consistent strokes in line with the style images while preserving more content details compared to the other methods. NST employs an iterative process to generate stylized outcomes from white noises, resulting in unremoved noise(4th and 5th rows) and discrepant color schemes(4th row). Owing to the oversimplified mean and variance matching, the stylized results of AdaIN display varying strokes from the given style(1st-3rd rows). WCT struggles with preserving content structure and Linear exhibits halo effects surrounding the primary content(3rd and 4th rows). AAMS exists dot-like artifacts and ReReVST is unsatisfactory for certain style images(3rd and 5th rows). AdaAttN fails to transfer essential colors and patterns in the given style images(3rd-5th rows). IECAST introduces eye-like artifacts(5th row) and shows inconsistent color schemes. AesPA-Net displays blurriness in its content(3rd rows). MCCNet and CSBNet can better balance style and content information but occasionally present grid-like artifacts.

## 5.3 ABLATION STUDIES

We conduct ablations on the critical components of our method with ADM backbone to verify their effectiveness on both conditional image generation and arbitrary style transfer. As shown in Tab. 3, the raw clean-estimation guidance technique exhibits a marked underperformance compared to the noised-sample-based classifier guidance on the image generation task. By incorporating the proposed first- and second-order momentum-based gradient filter, the sampling quality significantly improves. Additionally, the gradient suppression scheme further refines the sampling quality, by diminishing the impact of the unreliable early gradients.

Table 3: Ablation studies on two presented tasks, under different resolutions.

| Model | Image Gen.(256) | | Image Gen.(128) | | Image Gen.(64) | | Style Transfer.(256) | |
|---|---|---|---|---|---|---|---|---|
| | FID↓ | sFID↓ | FID↓ | sFID↓ | FID↓ | sFID↓ | $\mathcal{L}_{content}$ ↓ | $\mathcal{L}_{style}$ ↓ |
| Classifier Guidance(Dhariwal & Nichol, 2021) | 4.59 | 5.25 | 2.97 | **5.09** | 4.14[1] | 5.73[1] | - | - |
| Raw clean-estimation guidance | 4.99 | 5.58 | 3.15 | 5.58 | 7.04 | 12.73 | 5.09 | 1.58 |
| Raw + 1st Momentum | 4.93 | 5.46 | 3.10 | 5.55 | 4.42 | 8.07 | 4.91 | 1.53 |
| Raw + 2nd Momentum | 4.49 | 5.24 | 2.88 | 5.37 | 1.87 | 4.48 | 4.92 | 1.58 |
| Raw + Filtering(1st & 2nd Momentum) | 4.29 | 5.28 | 2.80 | 5.23 | 1.84 | 4.34 | 4.73 | **1.53** |
| **Raw + Filtering & Suppression** | **4.20** | **5.17** | **2.72** | 5.14 | **1.81** | **4.31** | **4.70** | 1.54 |

## 6 CONCLUSIONS

In this study, we seek to discover the key reason for the performance gap between clean-estimation based guidance and noised-sample based guidance, based on which we propose two targeted approaches, momentum-driven gradient filtering and gradient suppression, to improve the consistency and robustness of the clean guidance throughout the denoising process. In comparison to existing methods, our approach incurs no additional training cost while delivering superior performance. It also demonstrates considerable potential for various downstream tasks, offering a reduced training burden and a significantly expanded range of guidance types.

---

[1]We conduct the evaluation using the code from the official repository on the samples generated by the pre-trained DDPM and classifier, with the recommended parameters(Dhariwal & Nichol, 2021).

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

In the main context, we highlight key observations regarding the performance degradation of clean-estimation guidance. Consequently, we propose ***momentum-driven gradient filtering*** and ***gradient suppression*** to offer more consistent guidance throughout the denoising process and mitigate the negative effect of the detrimental guidance gradients at certain timesteps. To further clarify our proposed methods, we detail the implementation and present additional results for two downstream tasks in Sec. A and Sec. B, along with an additional discussion on our Momentum-driven Gradient Filtering approach in Sec. C.

# A  CLEAN GUIDED CONDITIONAL IMAGE GENERATION

## A.1  IMPLEMENTATION DETAILS

**Diffusion Backbone and Settings**  For ADM (Dhariwal & Nichol, 2021) backbone, we utilize the pretrained weights from its official repository (Prafulla et al.) on ImageNet, across all resolutions, as our diffusion model backbone. Specifically, this backbone incorporates the learnable variance scheme proposed by Nichol & Dhariwal (2021), along with a multi-resolution attention scheme and BigGAN's(Brock et al., 2018a) residual blocks for up/downsampling. All settings and hyperparameters are kept the same except that we finetune the guidance scale, as the optimal choice shifts when the gradient norm is modified. For DiT backbone, we also use the official pretrained weights for latent diffusion models and for the latent decoder. We manually disable the classifier-free guidance used by the DiT and take it as a pure conditional diffusion model. At each timesteps, we decode the latent code estimation $\widehat{z}_0$ into pixel space and apply the clean guidance on it. The hyperparameters of the momentum-driven gradient filtering are set as $\{\lambda : 0.001, \eta_m : 0.5, \eta_v : 0.75\}$ in all experiments without finetuning. We use relatively small $\eta_m$ and $\eta_v$ to ensure the sensitivity of the algorithm to the current frame. All metric results for this task are computed on 50K randomly generated samples using the evaluation script provided by Prafulla et al..

**Classifier**  In this work, we implement two types of clean classifiers to ensure fair and comprehensive comparisons with previous works that employ noised classifiers. For both ADM and DiT backbone, we reuse the same pretrained noised classifiers provided by Prafulla et al. with a fixed time-embedding input $t = 0$ as a clean classifier. The results are presented in Tab.A.5 and Tab.4. These noised classifiers employ half of the UNet as the encoder and are trained with 500K iterations and a batch size of 256 at $256^2$ resolution, 300K iterations and a batch size of 256 at $128^2$ resolution, and 300K iterations and a batch size of 1024 at $64^2$ resolution. Notably, we also trained a clean classifier at $256^2$ resolution from scratch, using the same architecture as the noised classifier. The clean classifier is trained with 150K iterations and a batch size of 16, resulting in a total of 2.4 million ($16 \times 1.5 \times 10^5$) clean samples seen during training, while the noised classifier requires 128 million ($256 \times 5 \times 10^5$) samples for training.

| Model | FID↓ | sFID↓ | Prec↑ | Rec↑ | Model | FID↓ | sFID↓ | Prec↑ | Rec↑ |
|---|---|---|---|---|---|---|---|---|---|
| BigGAN-deep (Brock et al., 2018a) | 6.02 | 7.18 | **0.86** | 0.35 | BigGAN-deep (Brock et al., 2018a) | 4.06 | 3.96 | 0.79 | 0.48 |
| LOGAN (Wu et al., 2019) | 3.36 | - | - | - | IDDPM (Wu et al., 2019) | 2.92 | **3.79** | 0.74 | 0.62 |
| ADM (Dhariwal & Nichol, 2021) | 5.91 | 5.09 | 0.70 | **0.65** | ADM (Dhariwal & Nichol, 2021) | 2.07 | 4.29 | 0.74 | **0.63** |
| ***ADM with noised guidance*** | | | | | ***ADM with noised guidance*** | | | | |
| ADM + noised classifier (Dhariwal & Nichol, 2021) | 2.97 | **5.09** | 0.78 | 0.49 | ADM + noised classifier (Dhariwal & Nichol, 2021) | 4.14 | 5.73 | **0.84** | 0.54 |
| ***ADM with clean guidance only*** | | | | | ***ADM with clean guidance only*** | | | | |
| ADM + Plug-and-Play (Graikos et al., 2022) | 99.45 | 65.38 | 0.21 | 0.26 | ADM + Plug-and-Play (Graikos et al., 2022) | 98.25 | 43.94 | 0.26 | 0.33 |
| **Ours** | **2.72** | 5.14 | 0.80 | 0.58 | **Ours** | **1.81** | 4.31 | 0.77 | 0.60 |

Table 4: Sample quality comparison with state-of-the-art generative models on ImageNet 128 × 128 (left) and 64 × 64 (right). All the diffusion models are sampled using 250 DDPM steps.

## A.2  EVALUATION METRICS

The Fréchet Inception Distance (FID), proposed by Heusel et al. (2017), offers a comprehensive assessment that balances the diversity and fidelity of generation by comparing distribution differences between the real and generated manifolds. sFID(Nash et al., 2021) can be viewed as a variant of FID that captures the spatial relationships of images. Precision aims to measure the proportion of generated images that fall within the real manifold, while recall calculates the proportion of real

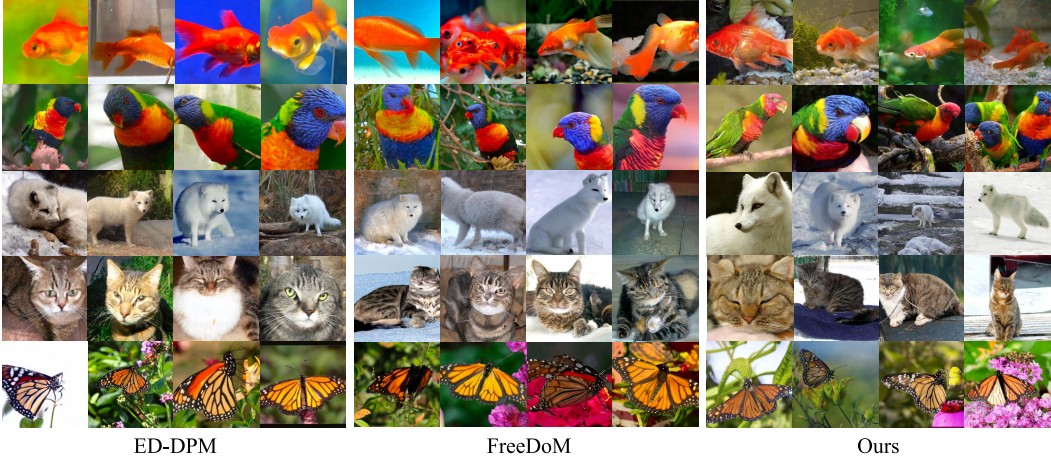

ED-DPM          FreeDoM          Ours

Figure 5: Qualitative comparison for the sampling results from $256 \times 256$ ADM model.

images falling into the generated manifold. Consequently, we use FID and sFID as primary evaluation metrics and precision and recall as secondary metrics to provide a comprehensive measurement.

### A.3 ADDITIONAL COMPARISONS

We present the comparison results for ADM at the resolutions of $128 \times 128$ and $64 \times 64$ in Table 4. We also provide a qulitative comparison with ED-DPM(Li et al., 2022b) and FreeDoM(Yu et al., 2023) at the resolution of $256 \times 256$ in Fig. 5 For a smaller resolution of $64^2$, we observed that guiding the DDPM with the pretrained noised classifier provided by Prafulla et al. does not enhance the sampling quality as it does for larger resolutions, such as $128^2$ and $256^2$. Employing only a clean classifier, our proposed method allows the clean-estimation guidance to not only surpass the performance of the noised guidance (Dhariwal & Nichol, 2021) but also achieve the best FID, while providing comparable sFID, Precision, and Recall.

### A.4 QUALITATIVE RESULTS

More qualitative results of different resolutions are provided in the Fig.6, Fig.7 and Fig.8.

### A.5 SAMPLING COST

We made a comparison on the sampling cost on $256 \times 256$ ADM model. The inference time is averaged on 10 denoising process running on an NVIDIA A100 GPU.

| Model | FID↓ | sFID↓ | inference time |
|---|---|---|---|
| ADM + raw clean guidance | 4.99 | 5.58 | 54.16 seconds/sample |
| ADM + FreeDoM | 8.66 | 6.84 | 259.95 seconds/sample |
| ADM + Plug-and-Play | 117.01 | 34.17 | 57.12 seconds/sample |
| ADM + ED-DPM | 5.98 | 5.93 | 54.39 seconds/sample |
| **ADM + Ours** | **4.20** | **5.17** | 54.20 seconds/sample |

## B ARBITRARY STYLE TRANSFER

In this part, we introduce the settings of our method for arbitrary style transfer in Sec. B.1. Also, we make a list of involved assets of this task in Sec. B.2.

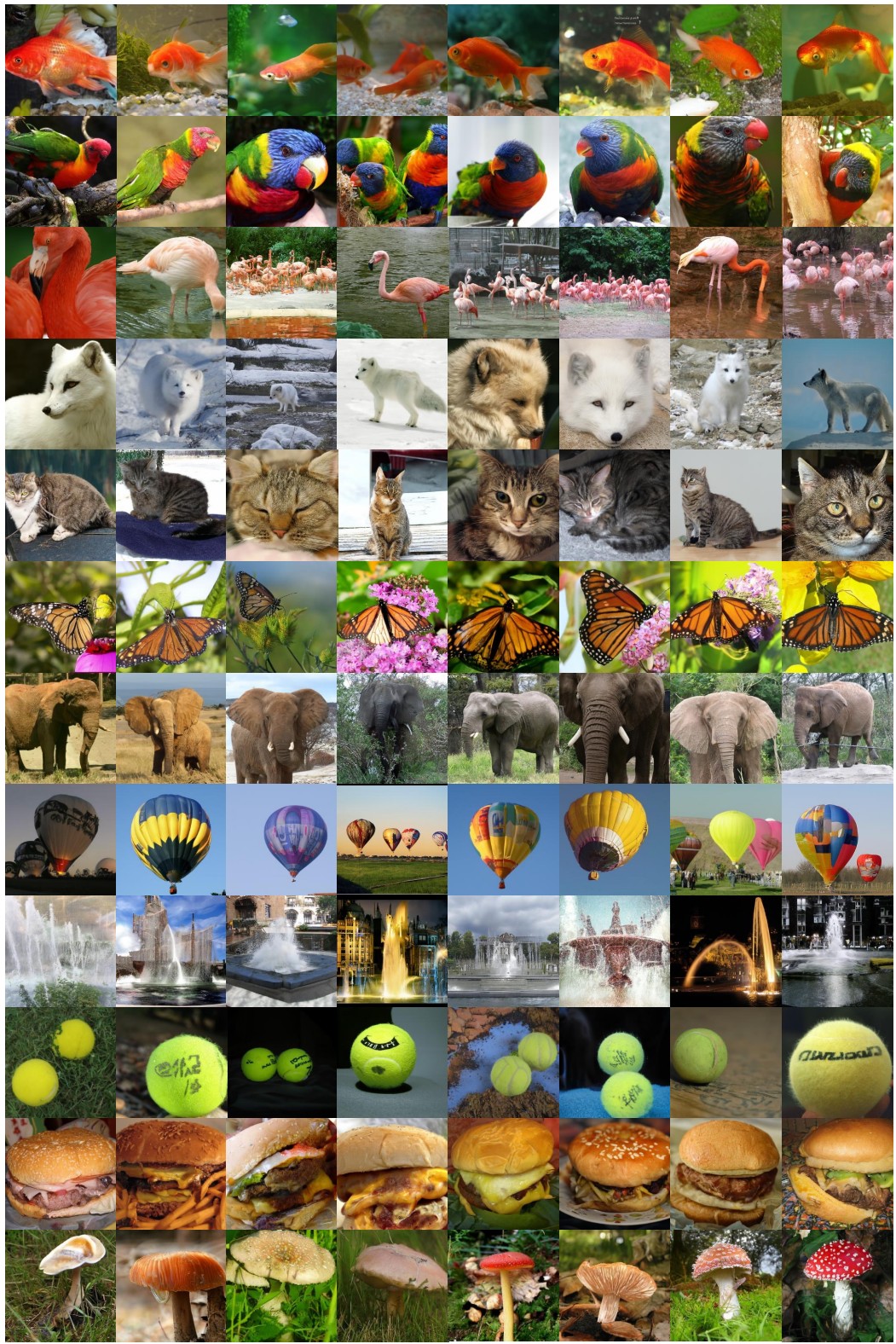

Figure 6: Qualitative results of our full pipeline at the resolution of $256 \times 256$. (FID: 4.20)

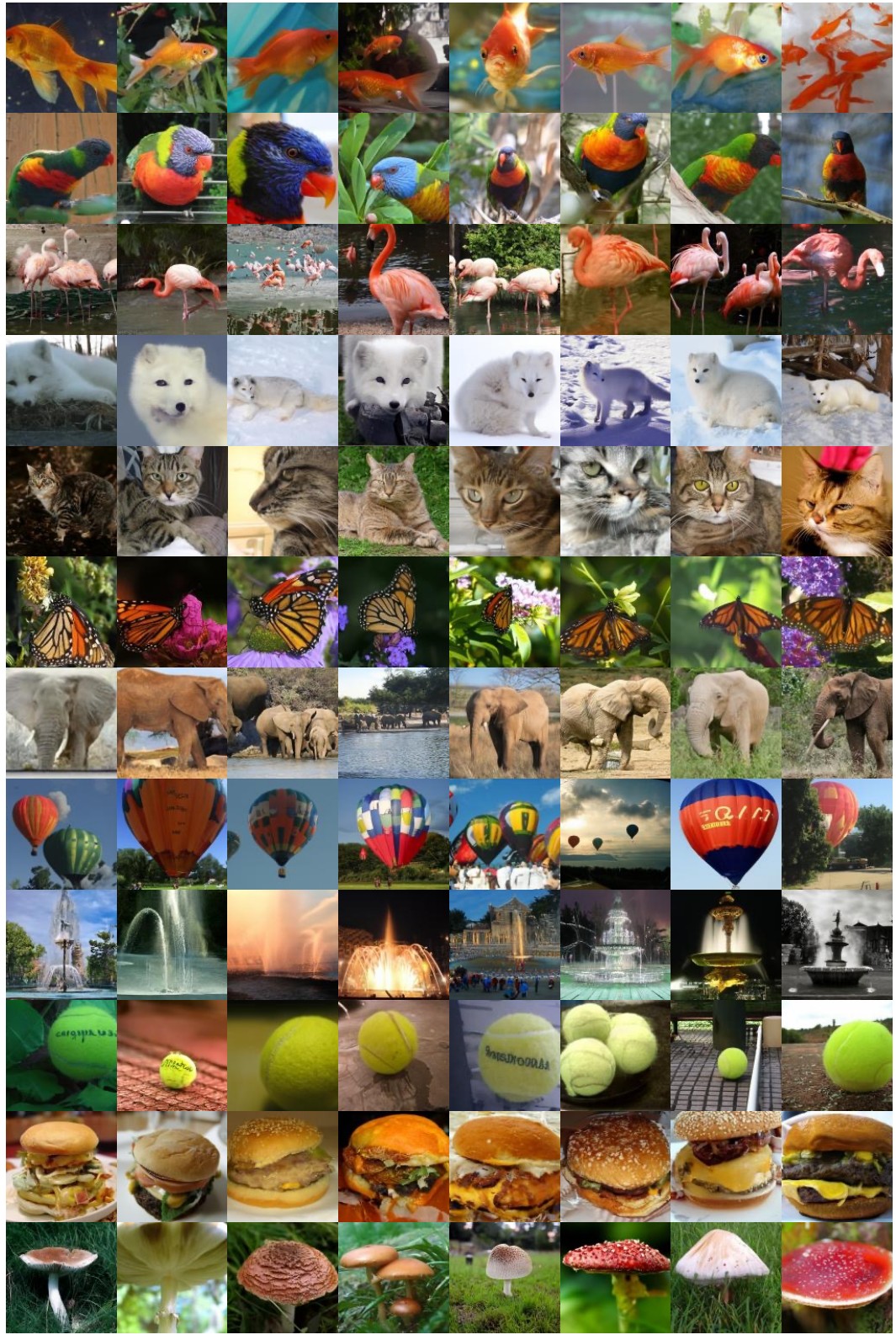

Figure 7: Qualitative results of our full pipeline at the resolution of $128 \times 128$. (FID: 2.72)

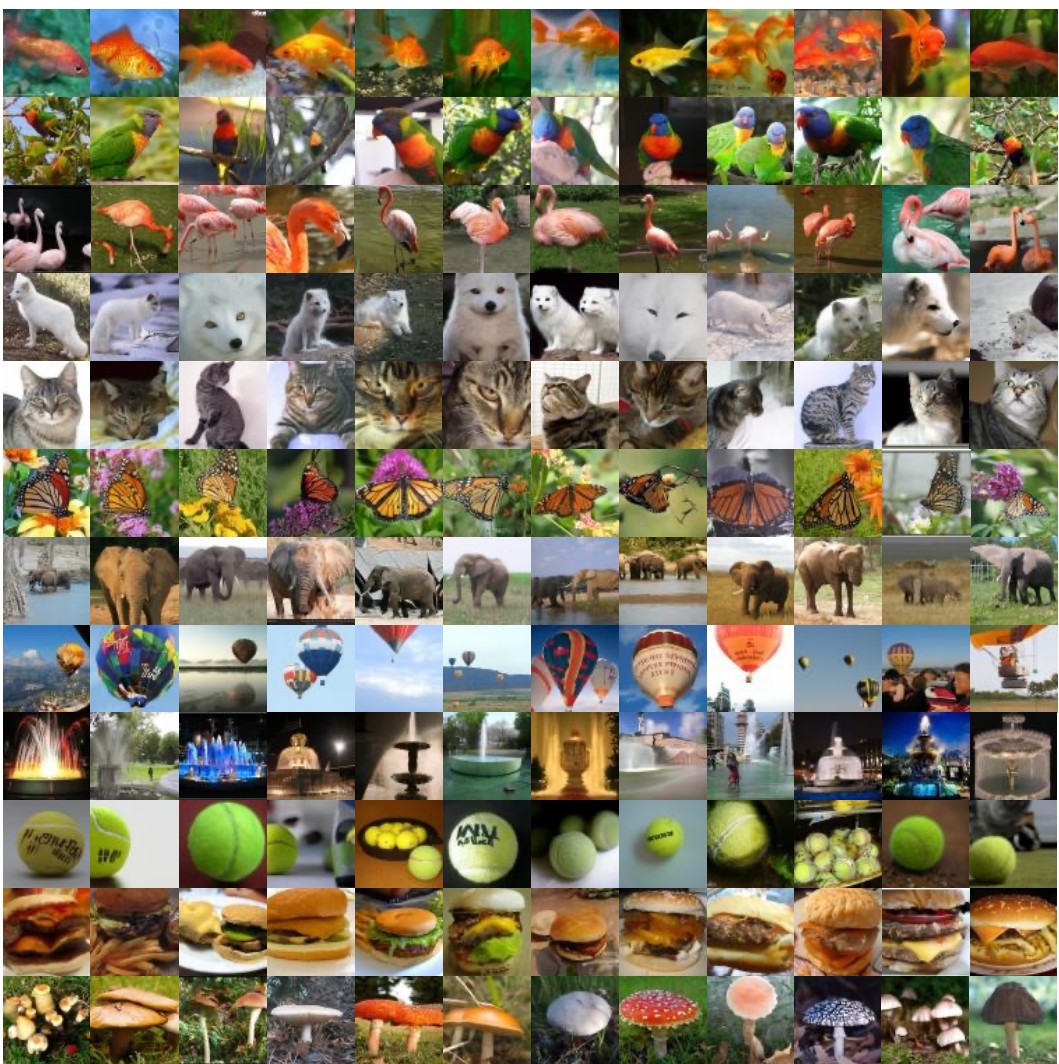

Figure 8: Qualitative results of our full pipeline at the resolution of $64 \times 64$. (FID: 1.81)

## B.1 ARCHITECTURE & SETTINGS

We trained an unconditional diffusion model on the WikiArt (Phillips & Mackintosh, 2011) dataset, with a learning rate of 0.0001 and a batch size of 8 for 1.2M iterations under the resolution of $256^2$. We still choose ADM (Dhariwal & Nichol, 2021) as the network architecture for the diffusion model, which adopts a linear noise schedule and unlearnable variance scheme. For content and style guidance, we use pretrained VGG-19 (Naoto) to obtain the encoded features of content and style images, where we use features from layer ReLU4_1 and ReLU5_1 for $\mathcal{L}_{content}$ computation and ReLU1_1, ReLU2_1, ReLU3_1, ReLU4_1 for $\mathcal{L}_{style}$ computation. $\lambda_c$ and $\lambda_s$ are set to 9000 and 18000 respectively.

## B.2 INVOLVED ASSETS

For the task of arbitrary style transfer, we compared our method with NST (Gatys et al., 2016), AdaIN (Huang & Belongie, 2017), WCT (Li et al., 2017b), Linear (Li et al., 2019), AAMS (Yao et al., 2019), MCCNet (Deng et al., 2021a), ReReVST (Wang et al., 2020), AdaAttN (Liu et al., 2021a), IECAST (Chen et al., 2021) and CSBNet (Lu & Wang, 2022). To make a strictly fair comparison with previous works, in this paper, we use their open source codes and follow their default settings for experiments. We use the WikiArt (Phillips & Mackintosh, 2011) dataset to train our diffusion model. Their URLs are reported as:

- NST: https://github.com/anishathalye/neural-style
- AdaIN: https://github.com/naoto0804/pytorch-AdaIN
- WCT: https://github.com/irasin/Pytorch_WCT
- Linear: https://github.com/sunshineatnoon/LinearStyleTransfer
- AAMS: https://github.com/JianqiangRen/AAMS
- MCCNet: https://github.com/diyiiyiii/MCCNet
- ReReVST: https://github.com/daooshee/ReReVST-Code
- AdaAttN: https://github.com/Huage001/AdaAttN
- IECAST: https://github.com/HalbertCH/IEContraAST
- CSBNet: https://github.com/Josh00-Lu/CSBNet
- WikiArt dataset: https://www.kaggle.com/competitions/painter-by-numbers/data?select=train.zip

## B.3 EVALUATION METRICS

We follow the same optimization target and evaluation metrics as those employed in previous works, such as (Deng et al., 2021b), (xin Zhang et al., 2022) and (Lu & Wang, 2022). Regarding recent learning-free methods (e.g., WCT), these approaches actively match the features of content images to style images by aligning the Gram matrices and minimizing their differences, while conscientiously preserving the original information of the content features through a whiten-like operation. Matching the Gram matrices amounts to optimizing the $\mathcal{L}_{style}$, as demonstrated by prior research (Li et al., 2017a). Additionally, the whiten operation, which aims to maintain content structures, corresponds to minimizing the $\mathcal{L}_{content}$.

## C FURTHER DISCUSSIONS ON MOMENTUM-DRIVEN GRADIENT FILTERING

### C.1 IMPLEMENTATION DETAILS

Inspired by the momentum-driven optimizers such as Adam(Kingma & Ba, 2014), our Momentum-driven Gradient Filtering strategies utilize the first- and second-order momentum to improve the consistency and robustness of the guidance gradient. We show the implementation details in Alg. 2 and Alg.3.

### C.2 FIRST-ORDER MOMENTUM

In Figure 2, we have illustrated how the first-order momentum can stabilize the guidance gradient. To further emphasize this effect, we present additional samples in Figure 9. Moreover, we highlight the

---

**Algorithm 2** Momentum-driven Gradient Filtering with only the $2^{nd}$ momentum.

---

**Input:** Raw gradients via clean-estimation guidance: $g_t$
**Input:** Learning rate: $\lambda$
**Input:** Exponential decay rates: $\eta_m, \eta_v$
**Initialize:** $v_{T+1} \leftarrow 0, \varepsilon = 10^{-8}$
**Yield:** Gradients (after gradient filtering): $\bar{g}_t$
    **for all** $t$ from $T$ to 1 **do**
        $v_t = \eta_v \cdot v_{t+1} + (1 - \eta_v) \cdot g_t^2$
        $\bar{v}_t = v_t/(1 - \eta_v^{T-t+1})$
        $\bar{g}_t = \lambda \cdot g_t/(\sqrt{\bar{v}_t} + \varepsilon)$
        **yield** $\bar{g}_t$
    **end for**

---

**Algorithm 3** Momentum-driven Gradient Filtering with both the $1^{st}$ and $2^{nd}$ momentum.

---

**Input:** Raw gradients via clean-estimation guidance: $g_t$
**Input:** Learning rate: $\lambda$
**Input:** Exponential decay rates: $\eta_m, \eta_v$
**Initialize:** $m_{T+1} \leftarrow 0, v_{T+1} \leftarrow 0, \varepsilon = 10^{-8}$
**Yield:** Gradients (after gradient filtering): $\bar{g}_t$
    **for all** $t$ from $T$ to 1 **do**
        $m_t = \eta_m \cdot m_{t+1} + (1 - \eta_m) \cdot g_t$
        $v_t = \eta_v \cdot v_{t+1} + (1 - \eta_v) \cdot g_t^2$
        $\bar{m}_t = m_t/(1 - \eta_m^{T-t+1})$
        $\bar{v}_t = v_t/(1 - \eta_v^{T-t+1})$
        $\bar{g}_t = \lambda \cdot \bar{m}_t/(\sqrt{\bar{v}_t} + \varepsilon)$
        **yield** $\bar{g}_t$
    **end for**

---

inconsistency and unreliability of clean estimations through a visualization of a few sub-sequences from their denoising process in Figure 10. Particularly, in the first and second examples, the clean estimations display inconsistent patterns(as marked by the red boxes) for the wings of the monarch and the head of the sandpiper. The estimations in the third row are too blurry, making it challenging for a clean classifier to provide a consistent and accurate prediction.

### C.3 SECOND-ORDER MOMENTUM

The second-order momentum adaptively assign a relatively small stepsize for the pivot pixels to make these pixels less vulnerable to the inaccurate gradients. We provide additional samples in Fig. 12 to demonstrate this effect. The figures reveal that foreground pixels, which are strongly related to the guidance condition, are frequently updated and accumulate a larger second-order momentum compared to background pixels during the denoising process. Consequently, the detrimental guidance gradients resulting from occasional incorrect clean estimations exert a reduced negative effect on the primary content within the generated samples.

Table 5: Sample quality comparison between the second-order momentum and a mere rescale on the gradient norm. All diffusion models are sampled using 250 DDPM steps.

| Model | FID↓ | sFID↓ | Prec↑ | Rec↑ |
|---|---|---|---|---|
| ADM + rescale | 5.26 | 5.72 | **0.86** | 0.45 |
| **ADM + $2^{nd}$** | **4.49** | **5.24** | 0.83 | **0.51** |

By assigning an adaptive stepsize for each pixel, the second-order momentum scales the gradient as a side effect, maintaining an approximately equal norm throughout the denoising process, as illustrated in Fig.11. As discussed in previous works(Li et al., 2022b), addressing the vanishing gradient issue in classifier guidance by uniformly scaling the gradient across the entire image can enhance sample quality for noised guidance. To validate the effectiveness of pixel-wise adaptive stepsize adjustment using the second momentum, we conducted an experiment that scales the raw

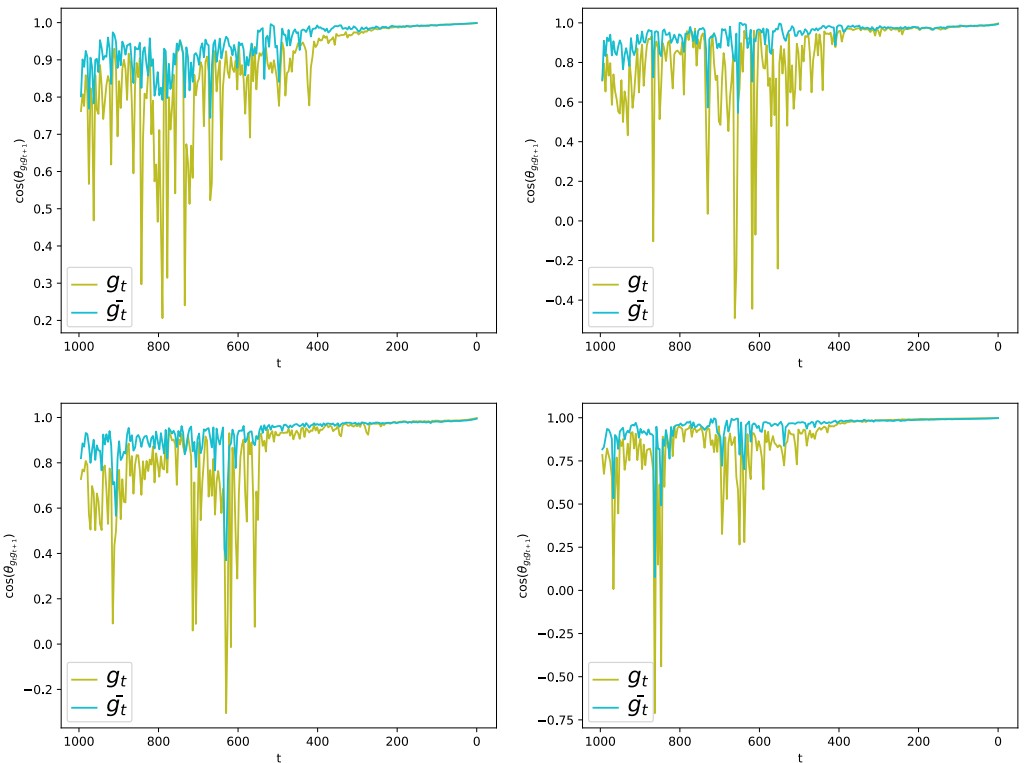

Figure 9: Additional random samples of the unprocessed gradient $g_t$ given by the clean-estimation guidance and the filtered gradient $\bar{g}_t$ by our proposed method. The cosine value of the angle between the gradient vector at time step $t$ and that at time step $t + 1$. The lower the curve, the noisier the gradient direction.

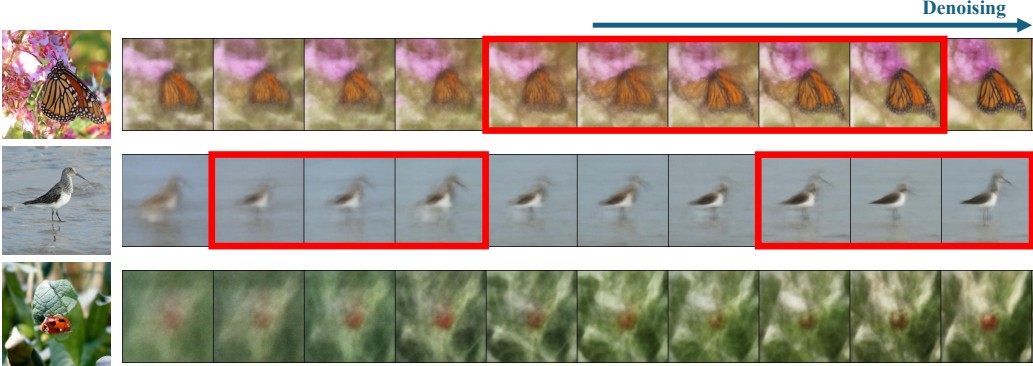

Figure 10: Random samples(left) and sub-sequences of their clean estimations(right) during the denoising process. Particularly, in the first and second examples, the clean estimations display inconsistent patterns(as marked by the red boxes) for the wings of the monarch and the head of the sandpiper. The estimations in the third row are too blurry, making it challenging for a clean classifier to provide a consistent and accurate prediction.

gradient norm to match the gradient filtered by the second-order momentum, based on the relative ratio of their L1-norms. The results in Tab.5 show that scaling the gradient with a uniform factor does not improve the sample quality for clean-estimation guidance. The reason for this is that it rescales the gradients of all pixels indiscriminately, which can lead to more severe contamination when the guidance gradient is inaccurate.

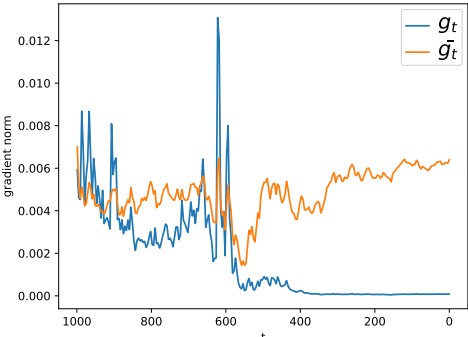

Figure 11: Gradient norm of the denoising process of a randomly generated sample, including the raw gradient via clean guidance $g_t$ and the gradient filtered by the second-order momentum $\bar{g}_t$. Both curves are multiplied by the classifier scale.

## D    THEORETICAL DISCUSSIONS

**Lemma 1.** *The clean estimation $\hat{x}_0(x_t, t)$ for a noised sample $x_t$ in discrete-time DPMs is actually the mathematical expectation $\mathbb{E}_{x_0 \sim q(x_0|x_t)}[x_0]$.*

***Proof.*** Revisiting the forward and reverse processes of diffusion models from the perspective of the score function, we have:

$$q(x_t|x_0) = \mathcal{N}(x_t|\alpha(t)x_0, \sigma^2(t)\mathbf{I}) \tag{18}$$

where $t \in [0, T]$, $\alpha(t)$ and $\sigma^2(t)$, (e.g. $\alpha(t) = \sqrt{\bar{\alpha}_t}$, $\sigma^2(t) = 1 - \bar{\alpha}_t$ in DDPM), determine the noise schedule of a diffusion model. Its corresponding differential equation (SDE) (Lu et al., 2022) is written as:

$$dx_t = f(t)x_t dt + g(t)d\omega_t \tag{19}$$

where $\omega_t$ is the standard Wiener process, and $f(t) = \frac{\mathrm{d}\log\alpha(t)}{\mathrm{d}t}$, $g_t^2 = \frac{\mathrm{d}\sigma^2(t)}{\mathrm{d}t} - 2\frac{\mathrm{d}\log\alpha(t)}{\mathrm{d}t}\sigma^2(t)$. Its equivalent reverse processLu et al. (2022) is:

$$dx_t = [f(t)x_t - g^2(t)\nabla_{x_t}\log q(x_t)]dt + g(t)d\bar{\omega}_t \tag{20}$$

where the only unknown term is the score function $s(x_t) = \nabla_{x_t}\log q(x_t)$. In discrete-time DPMs (e.g. DDPM), the negative scaled score function $-\sigma(t)s(x_t)$ is estimated using neural network $\epsilon_\theta(x_t, t)$ parameterized by $\theta$Lu et al. (2022). In general, the noise term $\epsilon_\theta(x_t, t)$ in discrete-time DPMs and score function $\nabla_{x_t}\log q(x_t)$ follows the equationBao et al. (2022b) that:

$$s(x_t) = \nabla_{x_t}\log q(x_t) = -\frac{\epsilon_\theta(x_t, t)}{\sigma(t)} \tag{21}$$

Further, $\mathbb{E}_{x_0 \sim q(x_0|x_t)}\nabla_{x_t}\log q(x_0|x_t) = \int \nabla_{x_t}q(x_0|x_t)dx_0 = \nabla_{x_t}\int q(x_0|x_t)dx_0 = 0$, we have:

$$\begin{aligned}
\nabla_{x_t}\log q(x_t) &= \nabla_{x_t}\log q(x_t) + \mathbb{E}_{x_0 \sim q(x_0|x_t)}\nabla_{x_t}\log q(x_0|x_t) \\
&= \mathbb{E}_{x_0 \sim q(x_0|x_t)}\nabla_{x_t}\log q(x_0, x_t) \\
&= \mathbb{E}_{x_0 \sim q(x_0|x_t)}\nabla_{x_t}\log q(x_t|x_0) \\
&= -\mathbb{E}_{x_0 \sim q(x_0|x_t)}\frac{x_t - \alpha(t)x_0}{\sigma^2(t)}
\end{aligned} \tag{22}$$

Accordingly, we further have:

$$\mathbb{E}_{x_0 \sim q(x_0|x_t)}[x_0] = \frac{1}{\alpha(t)}(x_t + \sigma^2(t)\nabla_{x_t}\log q(x_t)) \tag{23}$$

Substituting DDPM's noise schedule $\alpha(t) = \sqrt{\bar{\alpha}_t}$ and $\sigma^2(t) = 1 - \bar{\alpha}_t$ into Eq. 23, and combining with Eq. 21, we can get:

$$
\begin{aligned}
\mathbb{E}_{x_0 \sim q(x_0|x_t)}[x_0] &= \frac{1}{\alpha(t)}(x_t + \sigma^2(t)\nabla_{x_t}\log q(x_t)) \\
&= \frac{1}{\alpha(t)}(x_t - \sigma(t)\epsilon_\theta(x_t, t)) \\
&= \frac{1}{\sqrt{\bar{\alpha}_t}}(x_t - \sqrt{1 - \bar{\alpha}_t}\epsilon_\theta(x_t, t)) \\
&\triangleq \hat{x}_0(x_t, t)
\end{aligned}
\tag{24}
$$

**Remark 1.** *The second-order momentum enables the magnitude of the guidance update invariant to the scale of the gradient Kingma & Ba (2014); Qian (1999), thus helping normalize and regularize the scale of guidance gradients.*

**Remark 2.** *According to Lemma 1, directly calculating guidance gradients using the mathematical expectation $\mathbb{E}_{x_0 \sim q(x_0|x_t)}[x_0]$ brings unavoidable guidance gradient errors, which is caused by $|\mathbb{E}_{x_0 \sim q(x_0|x_t)}[x_0] - x_0|$.*

We formulate the gradient calculated using $\hat{x}_0(x_t, t)$ as $g_t$, the ground-truth gradient calculated using $x_0$ as $g_t^R$, and the normalized gradients using $2^{nd}$ momentum of time-step $t_0$ as $\hat{g}_t = g_t/(\sqrt{\bar{v}_{t_0}} + \epsilon)$ and $\hat{g}_t^R = g_t^R/(\sqrt{\bar{v}_{t_0}^R} + \epsilon)$, where $t \in [t_0, T]$, then we can formulate the error as:

$$
\hat{g}_t = \hat{g}_t^R + e_t
\tag{25}
$$

where $e_t$ represents the gradient error.

**Theorem 1.** *Assuming $\hat{g}_t = \hat{g}_t^R + e_t, e_t \in \mathcal{N}(0, \delta_t^2 \boldsymbol{I})$, $1^{st}$ momentum contribute to decreasing the gradient error and minimizing its variance after normalizing gradient using $2^{nd}$ momentum.*

**Proof.** According to the momentum update rules: $m_t = \eta_m m_{t+1} + (1 - \eta_m)g_t$, $v_t = \eta_n v_{t+1} + (1 - \eta_n)g_t^2$, $\bar{m}_t = m_t/(1 - \eta_m^{T-t+1})$, $\bar{m}_t = m_t/(1 - \eta_m^{T-t+1})$, $\bar{v}_t = v_t/(1 - \eta_n^{T-t+1})$, $\bar{g}_t = \lambda \cdot \bar{m}_t/(\sqrt{\bar{v}_t} + \varepsilon)$, and Eq 25, we can expand the processed gradient as:

$$
\bar{g}_t = \frac{\lambda(1 - \eta_m)}{1 - \eta_m^{T-t+1}} \sum_{i=t}^{T} \eta_m^{i-t} \frac{g_i}{\sqrt{\bar{v}_t} + \epsilon} = \frac{\lambda(1 - \eta_m)}{1 - \eta_m^{T-t+1}} \sum_{i=t}^{T} \eta_m^{i-t}(\hat{g}_i^R + e_i)
\tag{26}
$$

The gradient error using $2^{nd}$ momentum only is $E_t = \lambda e_t \sim \mathcal{N}(0, \lambda^2 \delta_t^2 \boldsymbol{I})$ and the gradient error using both $1^{st}$ and $2^{nd}$ momentum is $\bar{E}_t$, where $\bar{e}_t$ merges $T - t + 1$ Gaussians:

$$
\begin{aligned}
\bar{E}_t &= \frac{\lambda(1 - \eta_m)}{1 - \eta_m^{T-t+1}} \sum_{i=t}^{T} \eta_m^{i-t} e_i = \lambda \bar{e}_t \sim \mathcal{N}(0, \lambda^2 \bar{\delta}_t^2 \boldsymbol{I}) \\
\bar{\delta}_t^2 &= \frac{(1 - \eta_m)^2}{(1 - \eta_m^{T-t+1})^2} \sum_{i=t}^{T} \eta_m^{2i-2t} \delta_i^2
\end{aligned}
\tag{27}
$$

For the finite sequence $\{\delta_t^2\}_{t=0}^{T} = \{\delta_T^2, \delta_{T-1}^2, ..., \delta_0^2\}$, there exist an upper bound $P = \max_{i=0}^{T-1}\{\frac{\delta_{i+1}^2}{\delta_i^2}\}$, and $\bar{\delta}_t^2$ can be scaled up to:

$$
\bar{\delta}_t^2 \leq \frac{(1 - \eta_m)^2}{(1 - \eta_m^{T-t+1})^2} \sum_{i=t}^{T} \eta_m^{2i-2t} P^{i-t} \delta_t^2 \triangleq f(\eta_m)
\tag{28}
$$

Clearly, $f(\cdot)$ is a elementary function of $\eta_m$, and $f'(0) \triangleq \lim_{\eta_m \to 0^+} \frac{f(\eta_m) - f(0)}{\eta_m - 0} = -2\delta_t^2 < -\delta_t^2 < 0$, $f(0) \triangleq f(\eta_m)|_{\eta_m=0} = \delta_t^2 > 0$. According to the order-preserving properties of the limit, $\exists 0 < \zeta < 1, \forall 0 < \eta_m \leq \zeta, \frac{f(\eta_m) - f(0)}{\eta_m - 0} < -\delta_t^2 < 0$. Therefore, according to Eq. 28, we have:

$$
\bar{\delta}_t^2 \leq f(\eta_m) < f(0) = \delta_t^2, \qquad \forall 0 < \eta_m \leq \zeta
\tag{29}
$$

which demonstrates that for all $\eta_m \in (0, \zeta]$, error variance can be decreased. According to the extreme value theorem of continuous functions, there exists an **optimal** $\eta_m^* \in (0, \zeta]$ minimizing the error variance that serves as a tuning target for the hyper-parameter $\eta_m$.

# E  LIMITATIONS AND FUTURE WORKS

Although the current linear gradient suppression scheme demonstrates satisfactory performance in conditional image generation tasks, it does not adequately account for the actual quality of the guidance gradient. A more delicately designed suppression scheme might enhance the sampling quality further. For instance, we observed that suppressing the early gradient concerning the deviation of clean estimation yields superior performance for arbitrary style transfer. Moreover, the guidance functions employed for arbitrary style transfer are relatively rudimentary. Since our proposed methods permit arbitrary normalized objective functions as guidance, supplementary guidance conditions, such as the additional regularizer (Johnson et al., 2016b; Li & Wand, 2016), stroke loss (Jing et al., 2018), and MRFs constraints (Li & Wand, 2016), can be utilized concurrently to attain finer control over the stylized outcomes.

As a general approach for enhancing clean-estimation guidance, our proposed methods can be applied to improve the performance of numerous other clean-estimation-based techniques, such as those presented in (Fei et al., 2023). Furthermore, it is compelling to explore the application of our methods in various downstream tasks, including frame interpolation, novel view synthesis, and conditional 3D asset generation, accompanied by diverse guidance conditions.

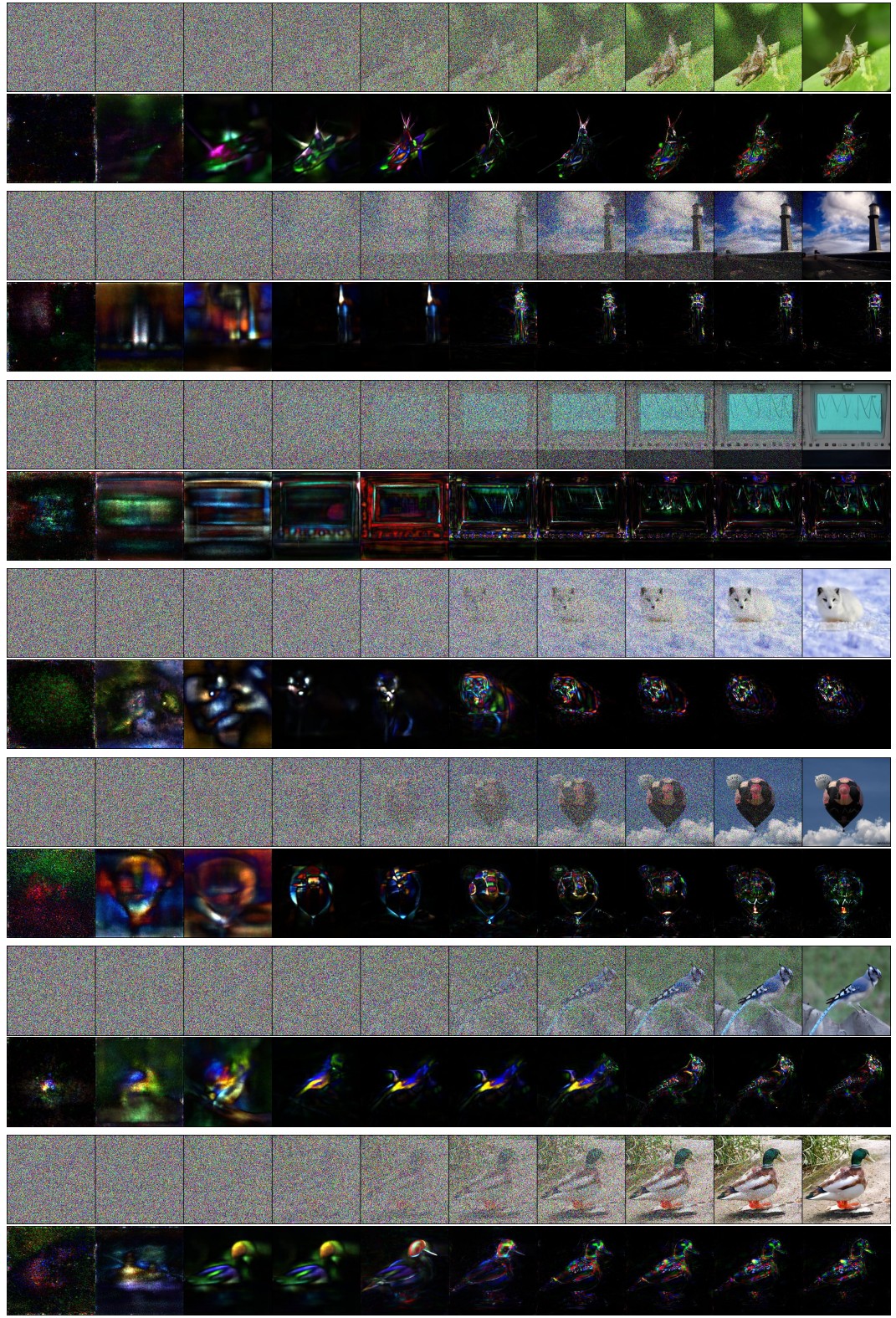

Figure 12: Additional random samples of the denoising process (upper) guided via clean-estimation and the corresponding second-order momentum (lower).

