# OpenReview forum: "Momentum-driven Noise-free Guided Conditional Sampling for Denoising Diffusion Probabilistic Models"
_ICLR.cc/2024/Conference — Submitted to ICLR 2024_

### Official Review · Reviewer_TEdQ · 2023-10-30

**Soundness:** 3 good
**Presentation:** 3 good
**Contribution:** 3 good
**Rating:** 6
**Confidence:** 4

**Summary:**

1.	The authors observe the unstable gradient in conditional sampling process. And the proposed momentum-driven gradient filtering algorithm to stabilize the conditional clean estimation gradient is sound, which uses the historical gradient to constrain the current gradient.
2.	The motivation of the gradient suppression is also reasonable. The prediction error is larger at the early stage of denoising process, so the corresponding gradient should be suppressed.

**Strengths:**

1.	The authors observe the unstable gradient in conditional sampling process. And the proposed momentum-driven gradient filtering algorithm to stabilize the conditional clean estimation gradient is sound, which uses the historical gradient to constrain the current gradient.
2.	The motivation of the gradient suppression is also reasonable. The prediction error is larger at the early stage of denoising process, so the corresponding gradient should be suppressed.

**Weaknesses:**

1.	Although the momentum gradient updating is reasonable, this may not be the first work to introduce momentum gradient updating to diffusion reverse sampling process. The work [1] has designed an adaptive momentum sampler to boost the performance, which also uses first-order and second-order. Since this work is also recent, so it is fine to me. Although that work focuses on unconditional sampling, I think maybe you should discuss the relation and difference between your work and that work.

[1] Boosting Diffusion Models with an Adaptive Momentum Sampler

2.	For the gradient suppression, I agree with the motivation that the direction error is larger at the early stage of the denoising process, and the corresponding sampling process should be adjusted. However, according to Algorithm1, the conditional gradient is suppressed. Since the semantic and layout is constructed when t is large, and the gradient compression is larger there, it may destroy the semantic matching between the final generated image and the condition. This may be more serious in some tasks with stricter requirement, such as inverse problems.

**Questions:**

See the weakness part

---

> ### Author Response · Authors · 2023-11-19
>
> Thanks for your constructive feedback. We highly value your comments and address them as follows:
>
> > Q1: The relation and difference between our work and the recent work *Boosting Diffusion Models with an Adaptive Momentum Sampler*
>
> The recent work titled *Boosting Diffusion Models with an Adaptive Momentum Sampler* leverages the momentum algorithm to ascertain the prior denoising direction from preceding timesteps in the context of an unconditional denoising process. Our work, however, takes a different path. We use the momentum algorithm to enhance the consistency and robustness of the guidance gradient during the conditional sampling process. Despite the similar functions of the first- and second-order momentum in smoothing the gradient direction and adaptively assigning the update step size in both studies, our motivation for using momentum algorithm and the problem we seek to solve diverge from the aforementioned work.
>
>
>
> > Q2: Since the semantic and layout is constructed when t is large, and the gradient compression is larger there, it may destroy the semantic matching between the final generated image and the condition.
>
> Thanks for this valuable questions. We think for tasks that requires much stricter guidance conditions, such as segmentation map, the problem of semantic and layout mismatching may happen due to the misalignment between the unconditional denoising process and the given conditions. For such kind of tasks, we think excluding the gradient of the intial steps from the suppression scheme, or increasing the number of denoising steps can help to alleviate the problem.

---

### Official Review · Reviewer_SuQ1 · 2023-10-31

**Soundness:** 2 fair
**Presentation:** 3 good
**Contribution:** 2 fair
**Rating:** 5
**Confidence:** 4

**Summary:**

This paper introduces an approach for conditional sampling in denoising diffusion probabilistic models (DDPM). It figures out that the performance gap between the clean and noised sample-based methods is mainly caused by the incorporation of estimation deviation in the clean-estimation process, and tries to solve it by implementing momentum-driven gradient filtering and a guidance suppression scheme. The proposed method exhibits superior performance in clean guided conditional image generation, and showcases its state-of-the-art capability in arbitrary style transfer tasks without the requirement of labeled datasets.

**Strengths:**

1. The author theoretically analysis the drawbacks of the clean guided methods and provide some results to proof their claims.
2. The author propose a method to solve the problem, and conduct experiments to its effectiveness.
3. Except the style transfer part, the motivation is clear and the paper is well-written.

**Weaknesses:**

1. About the style transfer task, I do not think it is appropriate to appear in this paper. According to my understanding, the goal of this paper is to bridging the performance gap between clean and noised sample-based methods. However, the experiment of style transfer task has no relationship with this topic. Therefore, I think the author better give an explanation about why conducting this part of experiments. Besides, according to the above reason, I also recommend the author to remove Section 2.3.
2. Lacking the results of other works on improving clean-estimation guidance in style transfer task.
3. The authors claim in Section 4.2 that the clean classifier+guidance function proves to be less reliable than the noise-robust classifier. Could the authors give a more direct proof like Figure 2?
4. About the experiments in Section 5.1:
1) the authors do not clarify which table is the results of this experiment
2) the authors mention FreeDoM and Universal Guidance in the paper, but only provide the results of ADM+FreeDoM. What about the results of DiT+FreeDoM and DiT/ADM+Universal Guidance?
5. The authors is recommended to prove the results of other methods on image generation task for comparison.
6. In Section 5.2, there is no description about what is I_cs.
7. In Section 5.3, the ablation results on style transfer task indicate that "Raw+Filtering & Suppression" can achieve the best results on L_content. However, it's results on L_style is only the third best. Does this indicate that "Raw+Filtering & Suppression" tends to preserve the original image's features rather than transferring to a new style?

**Questions:**

See weaknesses.

---

> ### Author Response · Authors · 2023-11-19
> **Response to Reviewer SuQ1 (1/2)**
>
> Thanks for your thoughtful reviews. We highly value your comments and address them as follows:
>
> > Q1: The experiment of style transfer task has no relationship with this topic. Therefore, I think the author better give an explanation about why conducting this part of experiments. Q2: Lacking the results of other works on improving clean-estimation guidance in style transfer task.
>
> We appreciate the reviewer's query about the inclusion of the style transfer task in our paper. While we understand the concern, the purpose of incorporating this experiment was two-fold.
> The first is that previous methods based on noised samples are not suitable for conditional generation tasks on unlabeled datasets, such as arbitrary style transfer. By achieving arbitrary style transfer, we underscore the potential and flexibility offered by our clean sample-based method.
> Secondly, the effectiveness of our proposed method is further demonstrated through comparisons and ablation studies on style-transfer task. For a more comprehensive analysis, we include a comparison with the results from FreeDoM as follows. To ensure a fair comparison, we set the total sampling steps for FreeDoM to be 1110 steps, incorporating its time-travel strategy, while our method uses 1000 ddpm steps for sampling. We find that results from FreeDoM contains dotted noise and shows a inferior sampling quality. As the results show, our method outperforms FreeDoM on both $ L_{content} $ and $ L_{style}$. The qualitative comparison will be updated later.
>
>  | Metrics  |  Ours  | FreeDoM  |
> | ------ |:------:|---------:|
> |$\mathcal{L}_{content}$|  4.70  |  6.30   |
> |$\mathcal{L}_{style}$  |  1.54  |  2.10   |
>
> > Q3: The authors claim in Section 4.2 that the clean classifier+guidance function proves to be less reliable than the noise-robust classifier. Could the authors give a more direct proof like Figure 2?
>
> In Appendix.C.2, we have supplemented our discussion with several sample sequnences of clean-estimations obtained during the denoising process in Figure 10. Upon examining these figures, it becomes evident that the clean-estimation is unstable during the denoising process(eg. in the red boxes). This instability subsequently results in an inconsistent guidance gradient throughout the denoising process.
>
> > Q4(1): The authors do not clarify which table is the results of this experiment.
>
> Thanks for pointing out this issue. We have made the necessary addition in the PDF, referencing `As shown in Tab. 1' in Section 5.1.
>
> > Q4(2): The authors mention FreeDoM and Universal Guidance in the paper, but only provide the results of ADM+FreeDoM.
>
> We have add the following comparison with DiT + FreeDoM here. We only provide the DiT + FreeDoM results here because FreeDoM is a follow-up work of Universal Guidance which shares a similar time-travel strategy and shows superior results. To ensure a fair comparison with our method, we have calibrated the total sampling steps for FreeDoM to closely align with ours. Our method yields superior outcomes on ADM model and style-transfer task, while demonstrates comparable results on DiT model. Given that DiT is a latent diffusion model, the guidance gradient $\frac{\partial \mathcal{P}}{\partial x_t}$ may exhibit a different pattern from ADM model. It may requires a futher tuning of our method's parameters.
>
> | Tables          | FID       |  sFID  | Precision | Recall |
> | --------------- |:---------:|:------:|:---------:|:------:|
> | DiT + raw clean |  3.54     |  5.22  |  0.80     |  0.56  |
> | DiT + FreeDoM   |  3.46     |  5.22  |  0.79     |  0.57  |
> | DiT + Ours      |  3.46     |  5.31  |  0.79     |   0.57 |
>
> > Q5: The authors is recommended to prove the results of other methods on image generation task for comparison.
>
> Thank you for your suggestion. To the best of our knowledge, we have compared our method with a variety of clean sample-based guidance approaches. However, further comparisons with noised sample-based methods might not provide a fair evaluation, given the inherent differences between these two kinds of methodologies. Clean sample-based methods normally yield inferior results than the noised sample-based methods, while they offer substantial flexibility in the choice of guidance functions and eliminate the additional training cost for noise-finetuning.

---

> ### Author Response · Authors · 2023-11-19
> **Response to Reviewer SuQ1 (2/2)**
>
> > Q6: In Section 5.2, there is no description about what is $I_{cs}$.
>
> Thank you for highlighting this oversight. We have now updated Section 5.2 to include the necessary explanation:  `The differences between the content image $I_c$ and the stylized image $I_{cs}$ is expressed as'.
>
>
> > Q7: In Section 5.3, the ablation results on style transfer task indicate that "Raw+Filtering \& Suppression" can achieve the best results on $ L_{content} $. However, it's results on $ L_{style} $ is only the third best.
>
> In Section 4.3, we provide an overarching insight that suppressing the early gradient can enhance the sampling quality. We employ the simplest linear scheme to demonstrate this effect. However, we acknowledge that more effective suppression schemes may exist. For instance, we discovered that the suppression scheme $\tilde{g_t} = \frac{\sqrt{\bar\alpha_t}}{\sqrt{1-\bar\alpha_t}}\bar{g_t}$ yields superior results for the style-transfer task, achieving both the lowest content error ($L_{content} = 4.69$) and the most optimal style error ($L_{style} = 1.46$). Moreover, we can readily strike a balance between the content error and style error by modulating the ratio of $\lambda_c:\lambda_s$.

---

### Official Review · Reviewer_hNt5 · 2023-10-31

**Soundness:** 3 good
**Presentation:** 3 good
**Contribution:** 2 fair
**Rating:** 5
**Confidence:** 3

**Summary:**

This paper focuses on conditional sampling of denoising diffusion probabilistic models with momentum-driven noise-free guidance. The proposed sampling method utilizes clean guidance functions, eliminating the need for additional training. Most importantly, they find that the primary reason for the worse performance of the previous noise-free guidance is by inaccurate clean estimations, especially during the early denoising stage. Then, they propose several simple yet effective methods for solving this problem. Furthermore, they demonstrate the feasibility and generalization capability on several downstream tasks like conditional image generation and style transfer.

**Strengths:**

1. The studied research topic DDPMs' conditional sampling is important and interesting in AIGC, which can be easily embedded into multiple DDPMs.
2. The authors found that the primary reason for the worse performance of the previous noise-free guidance is by inaccurate clean estimations, especially during the early denoising stage, which is also their main motivation.
3. The authors proposed the momentum-driven gradient filtering to filter noise in the gradient.
4. The experimental results on two downstream tasks including conditional generation and style transfer show that SOTA performance with many previous latest methods.

**Weaknesses:**

1. Limited Novelty: We know that momentum can obtain a more stable updating process, but as shown in Fig. 2b, the trends of filtered gradient and unprocessed gradient are almost the same rather than a gradual stabilization process. The simple method proposed in this paper is not the most effective solution to prevent noise in the gradient, but only alleviates the problem of early stage noise to some extent
2. Limited Performance Improvement: In Table 1 in Section 5.1, I note that the proposed method shows the limited improvement compared with 'raw clean guidance', e.g., DiT in sFID, Prec and Rec, and ADM in Prec and Rec.
3. Missed Experiments: 1) In Table 1 in Section 5.1, I note that the authors only compared clean estimation-based conditional without comparing noise-guidance-based methods. I am curious whether the results generated by the proposed method are comparable to noise-guidance-based methods. 2) In figures 5, 6, and 7 of the qualitative results in the appendix, I can't see any comparisons with other baselines. Please add some experiments to show its ability. 3) what is the specific sampling cost in the experiments? Please add experiments to compare clean-estimation guidance techniques.

**Questions:**

see weaknesses

---

> ### Author Response · Authors · 2023-11-19
>
> Thanks for your valuable feedback. We highly value your comments and address them as follows:
>
> > Q1: In Fig. 2b, the trends of filtered gradient and unprocessed gradient are almost the same rather than a gradual stabilization process. The simple method proposed in this paper is not the most effective solution to prevent noise in the gradient.
>
> Figure 2b illustrates the degree of deviation between the current gradient direction and that of the previous timestep. A smaller value on the graph signifies a greater deviation. As depicted in the graph, the filtered gradient presents a substantially more consistent direction. Notably, the significant deviations at certain timesteps (for instance, t=650, t=940) are markedly reduced. Moreover, the experimental results outlined in Table 1 further reinforce the effectiveness of our approach. It is clear that our method significantly enhances the performance of the clean-guided DDPM, surpassing other methods by a considerable margin, thereby establishing itself as the most effective solution to date.
>
>
> > Q2: In Table 1 in Section 5.1, I note that the proposed method shows the limited improvement compared with `raw clean guidance', e.g., DiT in sFID, Prec and Rec, and ADM in Prec and Rec.
>
> FID is a well-acknowledged and widely-used metric for evaluating image generation models, with sFID, Precision, and Recall typically serving as auxiliary metrics for a more comprehensive evaluation (Li et al., 2022b, Peebles & Xie, 2022). We wish to underscore that our method exhibits marked improvements on the ADM model for both FID and sFID metrics. Additionally, it demonstrates substantial enhancement on the DiT model for FID. These results collectively attest to the effectiveness of our proposed method.
>
> [1] Peebles, W. and Xie, S., 2023. Scalable diffusion models with transformers. In Proceedings of the IEEE/CVF International Conference on Computer Vision (pp. 4195-4205).
>
> [2] Zheng, G., Li, S., Wang, H., Yao, T., Chen, Y., Ding, S. and Li, X., 2022, October. Entropy-driven sampling and training scheme for conditional diffusion generation. In European Conference on Computer Vision (pp. 754-769). Cham: Springer Nature Switzerland.
>
> > Q3(1): In Table 1 in Section 5.1, I note that the authors only compared clean estimation-based conditional without comparing noise-guidance-based methods.
>
> We did not compare with noise-guidance-based methods because our proposed method is specifically focused on enhancing clean-estimation-based methods. Notably, our approach empowers clean-estimation guidance to surpass the performance of noise classifier guidance in terms of both FID (4.20 vs. 4.59) and sFID (5.17 vs. 5.25), while maintaining comparable levels of precision and recall.
>
> > Q3(2): In figures 5, 6, and 7 of the qualitative results in the appendix, I can't see any comparisons with other baselines. Please add some experiments to show its ability.
>
> We have add a qualitative comparison in Fig. 5 in Appendix.A.3.
>
> > Q3(3): What is the specific sampling cost in the experiments? Please add experiments to compare clean-estimation guidance techniques.
>
> We have add the following comparison on the sampling cost in Appendix.A.5.  The inference time is averaged on 10 denoising process running on an NVIDIA A100 GPU.
>
>
> |               Methods                |   FID  |  sFID|  Sampling Cost |
> |  ------------------------------ |:------:|:-------:|:---------------:|
> | ADM + raw clean guidance | 4.99  |  5.58   |  54.16  seconds/sample |
> | ADM + FreeDoM                |  8.66 |  6.84 |  259.95 seconds/sample  |
> | ADM + Plug-and-Play        | 117.01 | 34.17 | 57.12 seconds/sample |
> | ADM + ED-DPM                 | 5.98  |   5.93   |  54.39 seconds/sample  |
> | **ADM + Ours**              | **4.20** | **5.17** |	 54.20 seconds/sample  |

---

### Comment · Area_Chair_7rbY · 2023-11-21

Reviewers,

This is a reminder please reply to the rebuttal ASAP.

Please also post your final decisions according to the reviews, author responses and any paper revisions.

AC

---

### Meta-Review · Area_Chair_7rbY · 2023-12-09

**Metareview:**

The paper describes a new method for conditional sampling in denoising diffusion probabilistic models (DDPM) using noise-free guidance, improving guided image generation by stabilizing guidance gradients and reducing training costs, and demonstrating its versatility in tasks like arbitrary style transfer. Two reviewers recommend rejection and one recommends acceptance. The reviewers generally agree the paper motivation is clear and reasonable. however, the technical advancement provided by the paper is still limited and the image generation results are not strong enough. After carefully the rebuttal, reviews, and the paper, the AC agrees with the majority of the reviewers on rejecting the paper.

**Justification For Why Not Higher Score:**

The technical advancement provided by the paper is still limited and the image generation results are not strong enough.

**Justification For Why Not Lower Score:**

N/A.

---

### Decision · Program_Chairs · 2024-01-16

Reject